# RLAP-CLIP: Continual Multimodal Learning with Prototype Adaptation and Difficulty-Aware Routing

**Ruikun Luo**[1234], **Jiarui Wang**[1234], **Yuan Gao**[5]*, **Jing Yang**[5],
**Jieming Yang**[5], **Song Wu**[1234], **Hai Jin**[1234], **Xiaoyu Xia**[6]

[1]National Engineering Research Center for Big Data Technology and System
[2]Services Computing Technology and System Lab     [3]Cluster and Grid Computing Lab
[4]School of Computer Science and Technology, Huazhong University of Science and Technology
[5] Zhengzhou University
[6]Royal Melbourne Institute of Technology
{rkluo, m202574261, wusong, hjin}@hust.edu.cn,
{yuangaohnu, yangjinghust123, yjmlaile}@gmail.com,
xiaoyu.xia@rmit.edu.au

## Abstract

Vision-language models such as CLIP achieve strong zero-shot performance through contrastive pre-training but face significant challenges in class-incremental image classification scenarios. When learning new classes sequentially, current methods suffer from degradation in prototype quality due to passive averaging and underutilize their visual adaptation capabilities. We propose RLAP-CLIP, which addresses these limitations through three components. First, Reinforcement Learning-based Prototype Optimization (RLPO) formulates prototype construction as a reinforcement learning problem to actively optimize class separability rather than relying on simple averaging. Second, difficulty-aware cross-modal fusion uses a mixture-of-experts architecture to route samples through specialized processing pathways based on complexity. Third, dual-modal prompting balances visual and textual adaptation. Experiments on eight image classification benchmarks spanning general classification, fine-grained recognition, and domain-shift scenarios demonstrate consistent improvements, with RLAP-CLIP achieving average accuracy gains of up to 4.52 percentage points and final accuracy improvements of up to 6.26 percentage points over state-of-the-art methods.

## 1 Introduction

Vision-language models like CLIP Radford et al. (2021) have achieved remarkable zero-shot performance through large-scale contrastive pre-training, learning to align visual and textual representations in a shared semantic space. However, when deployed in continual learning scenarios where new tasks arrive sequentially, these models face the classic stability-plasticity dilemma Shi et al. (2024); Kim & Han (2023); Buzzega et al. (2020): they must acquire new knowledge while preserving previously learned concepts without catastrophic forgetting.

Current continual learning approaches for vision-language models predominantly rely on parameter-efficient adaptation He et al. (2023); Zhou et al. (2022b;a); He et al. (2025), freezing pre-trained weights while introducing minimal learnable parameters for new tasks. Methods like CoOp Zhou et al. (2022b) learn continuous prompts in the text encoder, while approaches such as MaPLe Khattak et al. (2023) extend prompting to both modalities. While this parameter-freezing strategy theoretically prevents catastrophic forgetting Zhou et al. (2023), our analysis reveals two critical limitations that prevent these methods from reaching their potential.

First, existing methods suffer from prototype quality degradation due to passive averaging. Since storing complete historical data violates memory constraints, continual learning methods rely on

---

*Corresponding author.

prototype-based knowledge retention Douillard et al. (2020), where each class is represented by the mean embedding of limited exemplar samples Zhou et al. (2025b); Zhu et al. (2021). However, as new tasks arrive and feature spaces evolve, prototypes computed through simple averaging may drift into suboptimal regions Masana et al. (2022), leading to progressive performance degradation and increased forgetting of previously learned classes Zhou et al. (2023).

Second, current approaches exhibit asymmetric multimodal exploitation, predominantly adapting textual representations while treating visual encoders as static feature extractors Zhou et al. (2022b); Wang et al. (2022); Jiang et al. (2025). This text-centric bias may break down in fine-grained recognition where discriminative visual patterns lack precise linguistic descriptors Song et al. (2020).

To address these challenges, we propose RLAP-CLIP, which targets the stability-plasticity dilemma through complementary mechanisms. *For stability*, we introduce RLPO, which transforms prototype construction from passive averaging into active optimization based on reinforcement learning. RLPO maintains robust class representations that resist degradation as new tasks arrive by learning to weight samples based on their contribution to class separability. *For plasticity*, we employ difficulty-aware cross-modal fusion with mixture-of-experts routing, enabling adaptive processing based on sample complexity, providing enhanced capacity for challenging boundary cases while efficiently handling easy samples. Bridging both objectives, our enhanced dual-modal prompting balances visual and textual adaptation, ensuring the model captures discriminative patterns across modalities without over-relying on either, thus maintaining flexibility for new tasks while preserving learned representations. Our main contributions are as follows:

- We identify and mitigate prototype quality degradation by formulating a reinforcement learning–based optimization framework;
- We propose a difficulty-aware sample routing strategy leveraging a mixture-of-experts architecture to enhance feature processing.
- We reveal the importance of balanced multimodal adaptation for continual learning in vision–language models.
- Experiments across eight benchmark datasets demonstrate consistent improvements, with RLAP-CLIP achieving average accuracy gains of up to 4.52 points and final accuracy improvements of up to 6.26 points over state-of-the-art methods, validating that RLAP-CLIP achieves state-of-the-art performance.

## 2 MOTIVATION

When deployed in continual learning scenarios, vision-language models show a significant performance gap compared to their theoretical potential Yi et al. (2024). Our empirical analysis using CLIP model Radford et al. (2021) reveals two critical issues that limit their effectiveness.

**Prototype Quality Deteriorates Over Sequential Tasks.** To overcome memory constraints, continual learning methods maintain class representations through prototypes Zhou et al. (2025b); Gomez-Villa et al. (2024), mean embeddings computed from a small set of stored exemplar samples per class. As Figure 1a demonstrates, this passive averaging strategy leads to substantial degradation across sequential tasks. While theoretical upper bounds computed using complete training datasets show only modest decline due to inherent feature space evolution, simple averaging exhibits severe performance drops and accelerating forgetting rates. The performance gap between uniform weighting and the theoretical optimum progressively widens as more tasks accumulate, revealing that passive averaging overlooks the need for active optimization based on class separability. This degradation becomes particularly problematic in continual learning scenarios where maintaining discriminative class boundaries is crucial for preventing catastrophic forgetting.

**Visual Adaptation Capabilities Remain Underutilized.** Current continual learning approaches exhibit asymmetric adaptation patterns: they primarily focus on textual prompts (learnable tokens that guide text encoder adaptation) Liu et al. (2025b); Kim et al. (2024); Smith et al. (2023); Zhou et al. (2022b) while treating visual encoders as fixed feature extractors. Visual prompts, which adapt the visual encoder through learnable tokens appended to image patches, remain largely unexplored despite their potential for capturing discriminative patterns that cannot be expressed linguistically. As shown in Figure 1b, When combining textual prompting with visual prompting(dual-modal prompting), performance increases markedly, 84.2% on CUB-200 and 66.9% on Aircraft. The t-SNE vi-

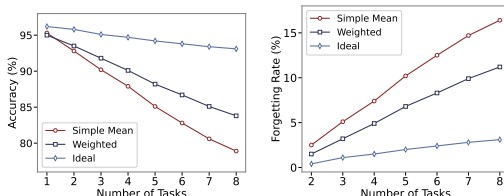
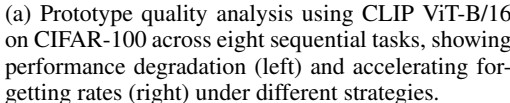
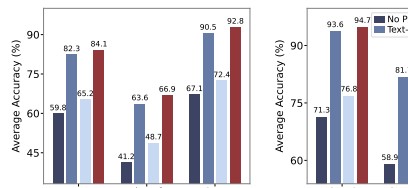

(a) Prototype quality analysis using CLIP ViT-B/16 on CIFAR-100 across eight sequential tasks, showing performance degradation (left) and accelerating forgetting rates (right) under different strategies.

(b) Performance comparison across different prompting strategies on six datasets, demonstrating that visual prompts provide substantial benefits to textual prompting

Figure 1: Comprehensive analysis demonstrating the importance of both prototype quality (a) and visual prompting strategies (b) in continual learning scenarios.

sualization in Figure 2 further confirms that dual-modal prompting produces more compact and well-separated class clusters, reducing overlap compared to single-modality approaches.

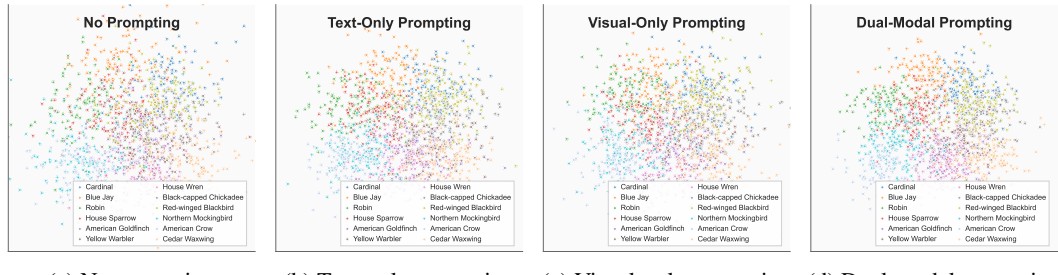

(a) No prompting  (b) Text-only prompting  (c) Visual-only prompting  (d) Dual-modal prompting

Figure 2: t-SNE visualization of learned feature representations on CUB-200 dataset under different prompting strategies. Colors represent 12 kinds of main bird species. Dual-modal prompting (d) achieves better class separation compared to no prompting (a), text-only (b), or visual-only (c).

To quantify the clustering improvements visualized in Figure 2, we measure intra-class compactness and inter-class separation across prompting strategies on CUB-200. Table 1 shows that dual-modal prompting achieves 32.2% tighter intra-class distance compared to no prompting while simultaneously increasing inter-class separation by 10.9%. The separation ratio, defined as the inter-class to intra-class distance ratio, improves from 1.90 to 3.11, representing a 63.7% enhancement. This quantitative analysis confirms that combining visual and textual adaptation produces significantly more discriminative feature representations than single-modality approaches, with dual-modal prompting outperforming both text-only (ratio: 2.74) and visual-only (ratio: 2.40) strategies.

These findings highlight that existing methods fail to exploit the potential of vision-language models fully. The reliance on passive prototype construction, coupled with asymmetric modality adaptation, results in a persistent performance bottleneck. Overcoming these challenges requires a rethinking of both knowledge representation and the effective utilization of multimodal capabilities in continuing learning.

Table 1: Quantitative clustering quality analysis on CUB-200 dataset. Lower intra-class distance indicates tighter clusters, higher inter-class distance indicates better separation, and higher separation ratio indicates superior overall clustering.

| Prompting Strategy | Intra-class Distance | Inter-class Distance | Separation Ratio |
|---|---|---|---|
| No Prompting | 2.08 | 3.95 | 1.90 |
| Text-Only Prompting | 1.56 | 4.28 | 2.74 |
| Visual-Only Prompting | 1.72 | 4.12 | 2.40 |
| Dual-Modal Prompting | **1.41** | **4.38** | **3.11** |

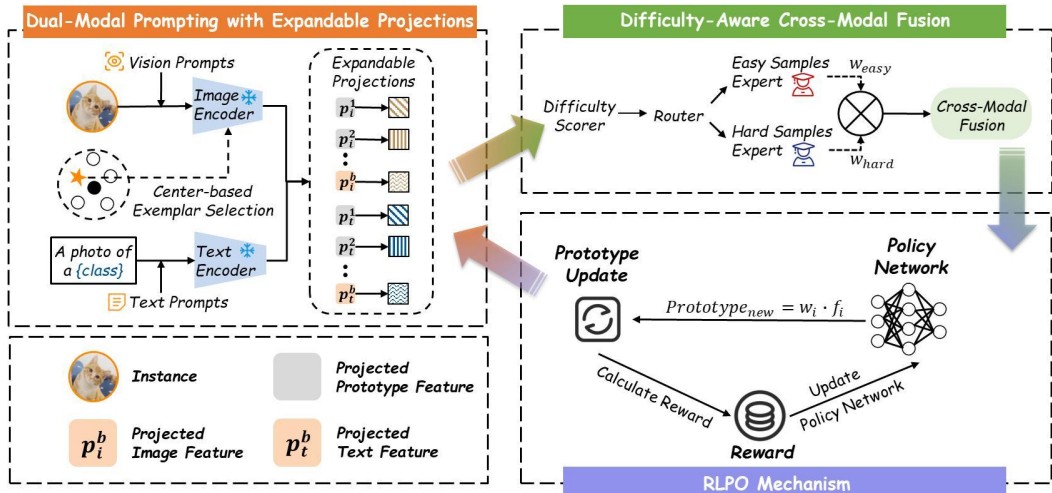

Figure 3: Overview of the RLAP-CLIP architecture for continual multimodal learning. It comprises enhanced dual-modal prompting with expandable projections, RLPO mechanism, and difficulty-aware cross-modal fusion.

## 3 RLAP-CLIP

To address these challenges, we propose **RLAP-CLIP**, a unified framework that integrates three key components: Reinforcement Learning-based Prototype Optimization (RLPO) to refine class representations and mitigate prototype degradation actively, difficulty-aware cross-modal fusion to dynamically route samples based on complexity, and enhanced dual-modal prompting to balance visual and textual adaptation. RLPO provides theoretical convergence guarantees in Appendix C.

### 3.1 REINFORCEMENT LEARNING-BASED PROTOTYPE OPTIMIZATION

Traditional prototype-based methods compute class representatives through uniform averaging: $p_c = \frac{1}{|\mathcal{E}_c|} \sum_{x_i \in \mathcal{E}_c} f(x_i)$, where $\mathcal{E}_c$ denotes the exemplar set for class $c$ and $f(\cdot)$ represents the feature extraction function. This approach treats all samples equally, leading to suboptimal prototypes when exemplars contain outliers or represent multi-modal distributions. We instead cast prototype construction as a sample-weighted optimization problem, where a policy network $\pi_\theta$ with parameters $\theta$ learns importance weights $w_i$ for each sample based on its contribution to class separability:

$$p_c = \sum_{i:y_i=c} w_i f_i, \quad w_i = \pi_\theta(f_i, \mathcal{P}) \quad \text{and} \quad \sum_{i:y_i=c} w_i = 1 \tag{1}$$

Here, $f_i$ denotes the feature embedding of sample $i$, $y_i$ is its class label, and $\mathcal{P} = \{p_1, \dots, p_C\}$ represents the set of all $C$ class prototypes. The policy network considers both individual sample features and global prototype relationships.

We define a reward function that encourages high intra-class similarity while maintaining inter-class separation. For sample $i$ with label $y_i$, the reward is:

$$R_i = \text{sim}(f_i, p_{y_i}) - \max_{j \neq y_i} \text{sim}(f_i, p_j) - \lambda \sum_{j \neq y_i} \frac{\text{sim}(p_{y_i}, p_j)}{C-1} \tag{2}$$

where $\text{sim}(\cdot, \cdot)$ denotes cosine similarity, $C$ is the total number of classes, and $\lambda$ controls the regularization strength. The first term maximizes similarity between sample $i$ and its class prototype $p_{y_i}$, the second term maximizes the margin to the nearest incorrect prototype, and the third term penalizes excessive similarity between different class prototypes to prevent collapse.

To stabilize training across tasks with varying reward scales, we normalize the rewards within each batch. Specifically, for a batch of samples with rewards $\{R_1, ..., R_B\}$, we compute normalized

advantages:

$$A_i = \frac{R_i - \mu_R}{\sigma_R + \epsilon} \tag{3}$$

where $\mu_R = \frac{1}{B}\sum_{i=1}^{B} R_i$ is the batch mean, $\sigma_R = \sqrt{\frac{1}{B}\sum_{i=1}^{B}(R_i - \mu_R)^2}$ is the batch standard deviation, and $\epsilon$ prevents division by zero.

The policy network $\pi_\theta$ is trained to maximize the expected reward while preventing drastic changes from a reference policy. The optimization objective is:

$$\mathcal{L}_{\text{RLPO}} = -\frac{1}{B}\sum_{i=1}^{B}\log \pi_\theta(w_i|f_i, \mathcal{P}) \cdot A_i + \lambda_{\text{KL}} D_{\text{KL}}(\pi_\theta \| \pi_{\text{ref}}) \tag{4}$$

where the first term encourages the policy to assign higher weights to samples with positive advantages (samples that improve class separation), and the second term is the Kullback-Leibler divergence:

$$D_{\text{KL}}(\pi_\theta \| \pi_{\text{ref}}) = \sum_{i=1}^{B} \pi_\theta(w_i|f_i, \mathcal{P}) \log \frac{\pi_\theta(w_i|f_i, \mathcal{P})}{\pi_{\text{ref}}(w_i|f_i, \mathcal{P})} \tag{5}$$

We implement $\pi_\theta$ as a four-layer MLP with dimensions [512→384→192→96→1], using Layer-Norm and GELU activation between layers. Output weights are normalized via softmax. The reference policy $\pi_{ref}$ is maintained through exponential moving average and updated after each task to prevent drastic policy shifts. The reference policy $\pi_{\text{ref}}$ is initialized as a uniform distribution (equal weights) and updated after each task to the learned policy, ensuring smooth adaptation across sequential tasks while preventing catastrophic changes.

## 3.2 DUAL-MODAL PROMPTING WITH EXPANDABLE PROJECTIONS

Existing methods predominantly adapt textual representations while treating visual encoders as fixed feature extractors Liu et al. (2025b); Kim et al. (2024), creating an asymmetry that hinders performance on fine-grained recognition tasks. We address this by introducing dual-modal prompting with task-specific projections to jointly adapt visual and textual features.

For each task $t$, we maintain learnable visual prompts $\mathcal{V}^t = \{v_1^t, v_2^t, ..., v_K^t\} \in \mathbb{R}^{K \times D}$ shared across all classes, where $K$ denotes the prompt length and $D$ is the embedding dimension. These prompts are appended to the patch embeddings of input image $x_i$, yielding the augmented visual input:

$$\tilde{x}_i = [x_i; \mathcal{V}^t] \tag{6}$$

where $[\cdot;\cdot]$ denotes concatenation. Similarly, we employ learnable textual prompts pretended to a fixed template. For class $c$, the textual input follows the template structure:

$$T_c = \mathcal{T}^t \oplus \text{"A photo of a [CLASS]"} \tag{7}$$

where $\mathcal{T}^t = \{t_1^t, t_2^t, ..., t_M^t\} \in \mathbb{R}^{M \times D}$ are learnable textual prompts with length $M$, and $\oplus$ denotes the concatenation operation in the token sequence. This design allows the learned prompts to provide context before the class-specific template, enabling better adaptation to task-specific characteristics.

To prevent interference between tasks while enabling adaptation, we employ task-specific projection layers that transform the frozen CLIP features:

$$f_i^{v,(t)} = P_v^t(E_v(\tilde{x}_i)), \quad f_c^{t,(t)} = P_t^t(E_t(T_c)) \tag{8}$$

where $E_v$ and $E_t$ denote the frozen visual and textual CLIP encoders respectively, $P_v^t : \mathbb{R}^D \to \mathbb{R}^{D'}$ and $P_t^t : \mathbb{R}^D \to \mathbb{R}^{D'}$ are task-specific linear projection layers that map features to a task-adapted space of dimension $D'$. By maintaining separate projection parameters for each task while keeping the backbone frozen, we preserve previously learned knowledge and avoid interference between tasks.

To maintain representative samples as the feature space evolves, we dynamically update exemplar sets for each class. Let $\mathcal{E}_c$ denote the current exemplar set and $\mathcal{S}_c$ the new samples for class $c$. We compute the centroid of the combined set:

$$\mu_c = \frac{1}{|\mathcal{E}_c| + |\mathcal{S}_c|}\left(\sum_{x_i \in \mathcal{E}_c} f^{v,(t)}(x_i) + \sum_{x_j \in \mathcal{S}_c} f^{v,(t)}(x_j)\right) \tag{9}$$

where $f^{v,(t)}(\cdot)$ represents the visual feature extraction through the encoder and projection. We then select the $m$ samples nearest to $\mu_c$ in the feature space to form the updated exemplar set $\mathcal{E}_c^{\text{new}}$:

$$\mathcal{E}_c^{\text{new}} = \underset{\mathcal{X} \subseteq \mathcal{E}_c \cup \mathcal{S}_c, |\mathcal{X}| = m}{\arg\min} \sum_{x \in \mathcal{X}} \|f^{v,(t)}(x) - \mu_c\|_2 \tag{10}$$

This exemplar selection strategy, which we term *center-based selection*, ensures that stored samples remain representative of their class as the feature space evolves across tasks. By prioritizing samples closest to the class centroid, we maintain exemplars that capture core discriminative characteristics while filtering outliers that could degrade prototype quality. This approach becomes particularly important when combined with RLPO, as the policy network relies on exemplar quality to learn effective sample weighting strategies.

### 3.3 DIFFICULTY-AWARE CROSS-MODAL FUSION WITH MIXTURE OF EXPERTS

Beyond prototype optimization, effective continual learning also requires adaptive processing that accounts for varying sample complexity within each task. While our dual-modal prompting captures discriminative patterns across modalities, not all samples benefit equally, straightforward examples may be over-processed while challenging boundary cases receive insufficient attention. To address this imbalance, we design a mixture-of-experts (MoE) mechanism that dynamically routes samples through specialized processing pathways based on their difficulty, providing the most benefit for maintaining class boundaries during continuing learning.

Firstly, we quantify sample difficulty using the distance from a sample to its corresponding class prototype. For sample $i$ with class label $y_i$, we define the difficulty score as:

$$d_i = 1 - \frac{\text{sim}(f_i^v, p_{y_i}^v) + \text{sim}(f_i^t, p_{y_i}^t)}{2} \tag{11}$$

where $\text{sim}(\cdot, \cdot)$ denotes cosine similarity, $f_i^v$ and $f_i^t$ are the visual and textual features of sample $i$ respectively, and $p_{y_i}^v$, $p_{y_i}^t$ are the visual and textual prototypes for class $y_i$ obtained from RLPO. Samples with high difficulty scores (far from prototypes) are considered hard, whereas low-difficulty samples are deemed easy. This metric leverages the optimized prototypes from RLPO to provide reliable difficulty estimates.

Based on this difficulty metric, we employ two specialized experts: a lightweight expert $E_{\text{easy}}$ implemented as a single linear layer for simple samples, and a deep expert $E_{\text{hard}}$ consisting of a three-layer feed-forward network for complex samples. The routing probability for sample $x_i$ to the easy expert is computed as:

$$P(E_{\text{easy}}|x_i) = \sigma(-\alpha(d_i - \tau)) \tag{12}$$

where $\sigma(\cdot)$ is the sigmoid function, $\tau \in [0, 1]$ is a learned threshold parameter that determines the difficulty boundary, and $\alpha > 0$ controls the sharpness of the routing decision. The probability of routing to the hard expert is $P(E_{\text{hard}}|x_i) = 1 - P(E_{\text{easy}}|x_i)$. The final visual features after expert processing combine outputs from both pathways:

$$f_i^{v,\text{expert}} = P(E_{\text{easy}}|x_i) \cdot E_{\text{easy}}(f_i^v) + P(E_{\text{hard}}|x_i) \cdot E_{\text{hard}}(f_i^v) \tag{13}$$

For cross-modal fusion, we employ an attention mechanism to adaptively weight the contributions from different modalities. We first concatenate the expert-processed visual features $f_i^{v,\text{expert}}$, the prototype of visual features $f_{\text{visual proto}}$, and the textual features $f_{\text{textual proto}}$:

$$h_i = [f_i^{v,\text{expert}}; f_{\text{visual proto}}; f_{\text{textual proto}}] \tag{14}$$

where $[\cdot; \cdot]$ denotes concatenation. The attention mechanism computes importance weights for each modality:

$$[W_a, W_b, W_c] = \text{SoftMax}(\text{Attention}(h_i)) \tag{15}$$

where the attention function is implemented as a feed-forward network with ReLU activation that maps the concatenated features to three scalar weights. The weighted features are then computed as:

$$\tilde{f}_i^v = W_a \cdot f_i^{v,\text{expert}}, \quad \tilde{f}_{\text{visual proto}} = W_b \cdot f_{\text{visual proto}}, \quad \tilde{f}_{\text{txetual proto}} = W_c \cdot f_{\text{textual proto}} \tag{16}$$

These weighted features $f_i^{v,\text{expert}}$, $\tilde{f}_{\text{visual proto}}$, and $\tilde{f}_{\text{txetual proto}}$ are used for computing similarities in the final classification, allowing the model to dynamically emphasize different information sources based on sample characteristics.

## 3.4 TRAINING AND INFERENCE

Our model freezes CLIP encoders throughout training, adapting only task-specific prompts ($V^t$, $T^t$), projection layers ($P_v^t$, $P_t^t$), the RLPO policy network ($\pi_\theta$), MoE experts ($E_{easy}$, $E_{hard}$), and cross-modal attention weights. During training, prompts augment encoder inputs, projections map features to task-adapted space, and difficulty scores route samples through MoE pathways. RLPO jointly optimizes prototype weights via our multi-objective loss (Appendix D). At inference, we classify samples by nearest prototype matching after identical feature processing.

## 4 EXPERIMENTS

We evaluate RLAP-CLIP on diverse continual multimodal learning tasks. The model builds on CLIP with ViT-B/16 as the visual encoder, using pre-trained weights from OpenAI and LAION-400M Radford et al. (2021). Each task is trained for 20 epochs per task using AdamW optimizer Khattak et al. (2023). Following standard continual learning principles, we maintain 20 exemplars per class for fair comparison with other methods Rebuffi et al. (2017).

**Datasets and Baselines.** Experiments are conducted on eight datasets spanning different visual domains. Their details can be found in Appendix A and Appendix B. General classification tasks include CIFAR-100 Krizhevsky et al. (2009) and ImageNet-R Hendrycks et al. (2021). Fine-grained recognition is evaluated on CUB-200 Wah et al. (2011), Aircraft Maji et al. (2013), and Cars Krause et al. (2013). Specialized domains are covered by Food101 Bossard et al. (2014), UCF101 Soomro et al. (2012), and ObjectNet Barbu et al. (2019). Tasks arrive sequentially, requiring the model to learn new classes while retaining knowledge of previously encountered ones. We compare against three kinds of methods: Classical continual learning approaches, including various finetune variants Zhai et al. (2022); Zhou et al. (2022b); Prompt-based methods such as CODA-Prompt Smith et al. (2023) and DAP Jung et al. (2023); and Vision-language continual learning methods, including SimpleCIL Zhou et al. (2025a), PROOF Zhou et al. (2025b), DKR Cui et al. (2024), and C-CLIP Liu et al. (2025a).

Table 2: Performance comparison across eight datasets. Results show average accuracy (Avg) and final accuracy (Final) as percentages.

| Methods | Exemplar | CIFAR-100 | | CUB-200 | | Cars | | Aircraft | | Food-101 | | UCF-101 | | ImageNet-R | | ObjectNet | |
|---|---|---|---|---|---|---|---|---|---|---|---|---|---|---|---|---|---|
| | | Avg | Final | Avg | Final | Avg | Final | Avg | Final | Avg | Final | Avg | Final | Avg | Final | Avg | Final |
| Finetune | × | 5.30 | 2.46 | 0.56 | 0.47 | 1.54 | 1.13 | 1.72 | 1.05 | 2.14 | 1.52 | 1.21 | 0.80 | 1.01 | 0.88 | 0.69 | 0.54 |
| Finetune CoOp Zhou et al. (2022b) | × | 41.23 | 24.12 | 24.03 | 10.14 | 37.40 | 20.87 | 13.05 | 7.77 | 33.13 | 18.67 | 42.02 | 24.74 | 54.20 | 39.77 | 16.21 | 6.82 |
| Finetune LiT Zhai et al. (2022) | × | 27.69 | 7.67 | 51.95 | 35.96 | 83.08 | 78.23 | 25.10 | 13.77 | 29.74 | 12.05 | 81.79 | 65.40 | 57.75 | 29.77 | 32.85 | 17.17 |
| SimpleCIL Zhou et al. (2025a) | × | 80.20 | 76.63 | 79.75 | 77.52 | 88.96 | 86.85 | 53.05 | 48.09 | 84.73 | 81.65 | 88.12 | 85.68 | 76.84 | 74.48 | 45.11 | 40.13 |
| CoOp Zhou et al. (2022b) | ✓ | 78.34 | 73.04 | 74.09 | 67.47 | 87.98 | 86.60 | 41.81 | 39.18 | 81.74 | 76.35 | 88.36 | 85.71 | 79.76 | 77.13 | 40.40 | 34.47 |
| PLOT Chen et al. (2022) | ✓ | 74.35 | 67.90 | 78.35 | 72.03 | 82.43 | 74.26 | 46.82 | 43.58 | 78.39 | 72.49 | 87.09 | 82.91 | 70.45 | 68.24 | 41.85 | 33.38 |
| CODA-Prompt Smith et al. (2023) | ✓ | 81.33 | 75.92 | 79.81 | 74.73 | 89.45 | 87.84 | 62.05 | 54.70 | 79.35 | 73.46 | 92.73 | 90.28 | 79.32 | 74.73 | 47.86 | 42.35 |
| DAP Jung et al. (2023) | ✓ | 76.57 | 59.92 | 75.39 | 74.09 | 84.63 | 83.15 | 41.45 | 28.56 | 81.68 | 78.38 | 87.64 | 85.68 | 77.23 | 74.37 | 42.47 | 32.95 |
| DKR Cui et al. (2024) | ✓ | 80.17 | 77.35 | 78.94 | 76.23 | 88.75 | 86.92 | 61.28 | 56.47 | 85.91 | 82.73 | 89.32 | 87.15 | 80.47 | 78.18 | 47.23 | 41.89 |
| PROOF Zhou et al. (2025b) | ✓ | 82.92 | 78.87 | 81.67 | 79.18 | 90.53 | 89.54 | 63.59 | 58.81 | 87.52 | 84.74 | 93.56 | 91.32 | 82.32 | 80.30 | 49.64 | 43.65 |
| C-CLIP Liu et al. (2025a) | ✓ | 81.75 | 78.92 | 82.14 | 79.83 | 92.18 | 90.45 | 65.73 | 62.15 | 87.08 | 84.21 | 91.85 | 89.67 | 83.15 | 81.06 | 51.37 | 47.82 |
| **RLAP-CLIP** | ✓ | **86.64** | **79.41** | **85.78** | **83.67** | **94.82** | **93.15** | **70.25** | **68.41** | **88.24** | **86.88** | **97.68** | **95.80** | **85.63** | **82.22** | **53.89** | **48.91** |

## 4.1 MAIN RESULTS

Table 2 summarizes the experimental results across all eight datasets, reporting both average accuracy (performance across all tasks) and final accuracy (performance on all classes after learning the final task). Traditional fine-tuning approaches suffer from severe catastrophic forgetting, with final accuracies ranging from 0.47% on CUB-200 to 2.46% on CIFAR-100, indicating that parameter-based adaptation alone is insufficient for continual learning in vision-language models. Among prompt-based methods, CODA-Prompt shows better performance across most datasets, reaching 90.28% final accuracy on UCF-101. However, these approaches struggle with fine-grained discrimination, as seen in CODA-Prompt's 54.70% final accuracy on Aircraft compared to 90.28% on UCF-101. Besides, vision-language continual learning methods achieve better overall performance. PROOF establishes strong baselines with 78.87% final accuracy on CIFAR-100 and 91.32% on UCF-101, while C-CLIP demonstrates competitive results, particularly on visual recognition tasks like Cars (90.45%) and ImageNet-R (81.06%). Our RLAP-CLIP consistently outperforms all baselines across datasets and metrics. On general classification tasks such as CIFAR-100, it achieves

86.64% average accuracy, improving over PROOF by 3.72 points. The improvements are more pronounced on fine-grained recognition: on CUB-200, RLAP-CLIP reaches 85.78% average accuracy compared to C-CLIP's 82.14%, representing a 3.64 point gain. Similar improvements appear on Cars (+2.64 points over C-CLIP) and Aircraft (+4.52 points over C-CLIP). On challenging datasets like ObjectNet, RLAP-CLIP achieves 53.89% average accuracy, outperforming C-CLIP by 2.52 points. The improvements across both average and final accuracy metrics indicate that RLAP-CLIP not only learns new tasks effectively but also maintains better long-term knowledge retention, effectively addressing the stability-plasticity trade-off in continual learning.

## 4.2 ABLATION STUDIES

**Component Analysis.** We evaluate each component's contribution through incremental ablation. The baseline (Base) uses PROOF Zhou et al. (2025b) as the starting configuration, consisting of frozen CLIP encoders with text-only prompting and simple prototype averaging. Table 3 reports the performance as components are progressively incorporated.

Table 3: Component-wise ablation study. Numbers in parentheses indicate gains over the previous configuration. CMF denotes cross-modal attention fusion.

| Dataset | Base | +VP | +VP+MoE | +VP+MoE+RLPO | RLAP-CLIP |
|---|---|---|---|---|---|
| CIFAR-100 | 82.92 | 83.21 (+0.29) | 84.05 (+0.84) | 85.47 (+1.42) | 86.64 (+1.17) |
| CUB-200 | 81.67 | 82.34 (+0.67) | 83.52 (+1.18) | 84.89 (+1.37) | 85.78 (+0.89) |
| Cars | 90.53 | 91.28 (+0.75) | 92.67 (+1.39) | 93.94 (+1.27) | 94.82 (+0.88) |
| Aircraft | 63.59 | 64.83 (+1.24) | 66.45 (+1.62) | 68.73 (+2.28) | 70.25 (+1.52) |
| Food-101 | 87.52 | 87.68 (+0.16) | 87.84 (+0.16) | 87.95 (+0.11) | 88.24 (+0.29) |
| UCF-101 | 93.56 | 94.12 (+0.56) | 95.38 (+1.26) | 96.84 (+1.46) | 97.68 (+0.84) |
| ImageNet-R | 82.32 | 82.73 (+0.41) | 83.94 (+1.21) | 84.85 (+0.91) | 85.63 (+0.78) |
| ObjectNet | 49.64 | 50.89 (+1.25) | 52.47 (+1.58) | 53.21 (+0.74) | 53.89 (+0.68) |
| **Average** | 78.97 | 79.64 (+0.67) | 80.79 (+1.15) | 81.98 (+1.19) | 82.87 (+0.89) |

sively incorporated. Visual Prompts (VP) provide an average gain of 0.67%, with larger improvements on fine-grained datasets such as Aircraft (+1.24%) and smaller gains on coarse-grained tasks like Food-101 (+0.16%), indicating that VP captures discriminative patterns crucial for fine-grained recognition. The Mixture-of-Experts (MoE) routing mechanism contributes an additional 1.15% average improvement, with the largest gain on ObjectNet (+1.58%), by routing hard samples through deeper experts while efficiently processing easy ones to enhance feature quality. RLPO yields a further contribution (+1.19%), particularly on fine-grained tasks such as Aircraft (+2.28%) and action recognition like UCF-101 (+1.46%), by actively optimizing prototypes to preserve subtle class boundaries. Finally, cross-modal attention fusion (CMF), which adaptively weights visual, prototype, and textual features through the learned attention mechanism, provides an additional +0.89% on average, with the largest gains on Aircraft (+1.52%) and CIFAR-100 (+1.17%). The full RLAP-CLIP model, incorporating all four components, achieves consistent improvements across all datasets, confirming that each component contributes complementary benefits.

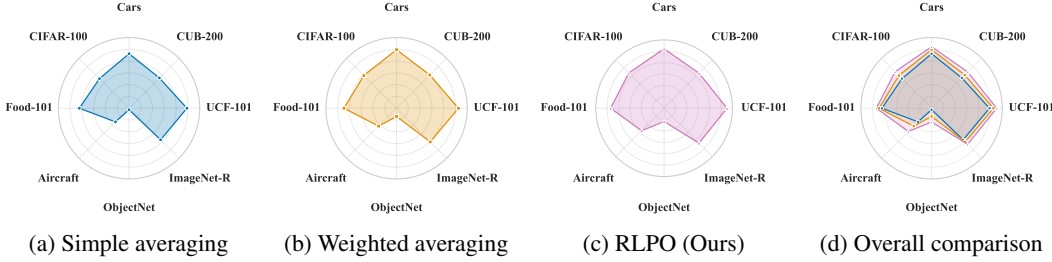

| (a) Simple averaging | (b) Weighted averaging | (c) RLPO (Ours) | (d) Overall comparison |

Figure 4: Comparison of prototype construction strategies. Performance ranges from 40% (center) to 100% (outer edge). Larger areas indicate better performance across datasets.

**Prototype Construction Strategies.** Figure 4 compares three prototype construction approaches across eight datasets. Simple averaging assigns equal weights to all exemplars, providing a basic baseline but yielding limited performance on ObjectNet (41.27%) and Aircraft (56.32%). Quality-weighted averaging refines prototypes by assigning higher weights to exemplars closer to the class centroid, assuming they better represent the class, leading to improved results (CUB-200: 79.86%, Aircraft: 61.48%). Our RLPO method goes further by learning weights through reinforcement learning. The policy network adaptively adjusts weights to maximize both intra-class cohesion and inter-class separation, achieving 83.42% on CUB-200, 92.18% on Cars, and 67.85% on Aircraft. As shown in Figure 4d, RLPO consistently outperforms passive averaging across all task types,

demonstrating its ability to adapt weighting strategies to task-specific characteristics and effectively mitigate prototype degradation.

RLAP-CLIP maintains reasonable computational efficiency with 8.5M trainable parameters and 41.7-minute training time per task, achieving a favorable balance between performance and cost (Appendix I). Our dual-modal prompting strategy outperforms parameter-efficient alternatives including LoRA by 10+ points (Appendix F), though visual prompting exhibits task-dependent trade-offs on fine-grained variants (Appendix G). Additional analyses in the appendix address practical deployment considerations. Appendix H examines performance under varying memory constraints (30/20/10/5/0 exemplars per class), demonstrating robust performance with graceful degradation. Appendix J presents t-SNE visualizations on Stanford Cars dataset, illustrating how RLPO achieves superior prototype alignment compared to simple mean averaging.

### 4.3 Hyperparameter Sensitivity Analysis

We analyze four key hyperparameters to understand their impact on model performance.

**RLPO Weight** ($\lambda_{RLPO}$). This parameter controls the influence of reinforcement learning-based prototype optimization in the total loss (Eq. 4). Figure 5a shows that all datasets achieve optimal performance at $\lambda_{RLPO} = 0.2$, with sharp degradation at higher values. UCF-101 drops from 97.68% to 92.15% as $\lambda_{RLPO}$ increases to 1.0, while fine-grained datasets like CUB-200 decline by 5.44 points. This uniform sensitivity suggests that excessive prototype optimization disrupts the learned feature space regardless of task type, with moderate reinforcement providing the best balance between class separation and feature stability.

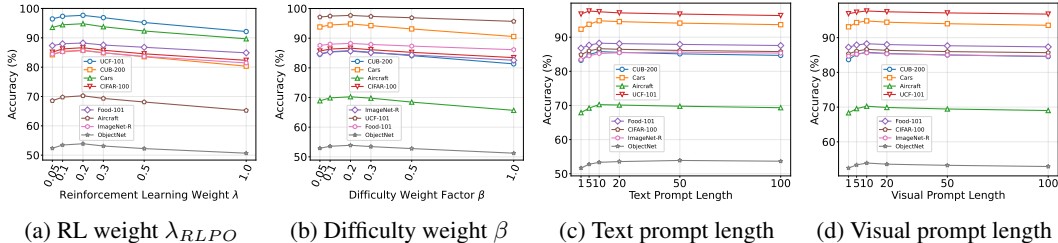

| (a) RL weight $\lambda_{RLPO}$ | (b) Difficulty weight $\beta$ | (c) Text prompt length | (d) Visual prompt length |

Figure 5: Hyperparameter sensitivity analysis across eight datasets. Optimal values vary by task characteristics but show consistent patterns within task categories.

**Difficulty Weight** ($\beta$). This factor amplifies the loss contribution of hard samples. Figure 5b reveals a consistent optimum at $\beta = 0.2$ across all datasets. Fine-grained recognition tasks exhibit higher sensitivity, CUB-200 drops 4.44 points from peak to $\beta = 1.0$, while Aircraft declines by 4.58 points. Hign difficulty weighting causes overfitting to boundary cases and outliers, particularly detrimental for tasks requiring subtle discrimination between visually similar classes.

**Text Prompt Length and Visual Prompt Length .** Text prompts (Figure 5c) show task-dependent optima, action recognition (UCF-101) peaks at 5 tokens, as action concepts require minimal linguistic elaboration. Fine-grained tasks optimize at 10 tokens, while distribution-shift datasets (ImageNet-R, ObjectNet) achieve best performance at 50 tokens. This pattern reflects varying linguistic complexity, where actions require simple descriptors, fine-grained categories need moderate detail, and distribution shifts benefit from extensive contextual guidance. Visual prompts (Figure 5d) uniformly optimize at 10 tokens across all tasks, with fine-grained datasets showing strongest gains (Aircraft: +1.91 points from 1 to 10 tokens). This consistency indicates that 10 visual tokens provide sufficient adaptation capacity without overfitting, particularly crucial given limited exemplar storage in continual learning.

## 5 Related Work

**Continual Learning in Vision-Language Models.** Recent approaches to continual learning in vision-language models primarily adopt parameter-efficient strategies to mitigate catastrophic forgetting Wang et al. (2023); Ostapenko et al. (2022). CoOp Zhou et al. (2022b) introduces learnable

continuous prompts for the text encoder while keeping CLIP backbone frozen. MaPLe Khattak et al. (2023) extends this to both visual and textual modalities with multi-modal prompting. CODA-Prompt Smith et al. (2023) develops an attention-based prompt pool that dynamically composes task-specific prompts, while DAP Jung et al. (2023) generates instance-level prompts for improved adaptability. SimpleCIL Zhou et al. (2025a) demonstrates that frozen CLIP features can achieve reasonable performance without adaptation, leading to methods like C-CLIP Liu et al. (2025a) and PROOF Zhou et al. (2025b) that combine prompt learning with prototype-based classification.

**Prototype Construction and Management.** Prototype-based approaches have been critical and common to continual learning, with iCaRL Rebuffi et al. (2017) pioneering the use of class mean embeddings computed from stored exemplars. Subsequent works improve exemplar selection Rebuffi et al. (2017) or correct representation bias Zhu et al. (2021), but continue treating prototype construction as a passive averaging process. PODNet Douillard et al. (2020) uses pooled outputs for distillation while maintaining prototypes for classification. Recent vision-language methods like PROOF Zhou et al. (2025b) and DKR Cui et al. (2024) adopt prototype-based classification but still compute prototypes through uniform averaging. This passive construction becomes problematic as feature spaces evolve across tasks Masana et al. (2022), leading to prototype drift and degraded performance. We reformulate prototype construction as an active optimization problem using reinforcement learning, where sample weights are learned to maximize class separability rather than assigned uniformly.

**Mixture of Experts and Adaptive Processing.** Mixture of Experts (MoE) architectures demonstrate that routing inputs through specialized pathways improves both efficiency and performance Fedus et al. (2022); Zhou et al. (2022c). Recent work shows MoE's effectiveness in large language models Dai et al. (2024); Shao et al. (2024) and multimodal settings Li et al. (2025); Pióro et al. (2024). In computer vision, adaptive processing has been explored through dynamic networks that adjust computation based on input complexity. However, these ideas have not been applied to continual learning in vision-language models, where samples vary significantly in difficulty both within and across tasks. We introduce difficulty-aware routing that allocates deeper processing to challenging samples near class boundaries while efficiently handling straightforward examples, addressing the varying complexity inherent in sequential task learning.

## 6 DISCUSSION AND CONCLUSION

We present RLAP-CLIP, which reformulates prototype construction as an active optimization problem using reinforcement learning rather than passive averaging. Experiments on eight datasets demonstrate consistent improvements, with the apparent improvements observed on fine-grained recognition tasks where visual adaptation is most impactful. Our results highlight three key findings: (1) actively optimized prototypes preserve class separability more effectively than uniform averaging, (2) dual-modal adaptation outperforms text-centric approaches, with visual prompting contributing substantial gains, and (3) difficulty-aware mixture-of-experts routing efficiently allocates computational resources while enhancing feature quality. While our method is evaluated under class-incremental settings, the varying improvements across task types suggest that the benefits of visual prompting are task-dependent. Extending RLAP-CLIP to other incremental learning paradigms and further exploring when visual adaptation offers the greatest benefit remain promising directions for future work.

## ACKNOWLEDGMENTS

We sincerely thank the AC and reviewers for their constructive and valuable feedback. This research was supported in part by the National Science Foundation of China (NSFC) under grant No. 62302130 and 62402188, the China Postdoctoral Science Foundation under Grant No. 2025T180424, the Postdoctoral Fellowship Program of CPSF under Grant No. GZC20240542 and GZC20251050, and the Postdoctoral Project of Hubei Province under Grant No. 2024HBB-HCXA026.

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

## APPENDIX

For a better understanding of the main paper, we provide additional details in this supplementary material, which is organized as follows:

**§A - Dataset**: Detailed specifications of the eight evaluation datasets, categorized into general classification tasks (CIFAR-100, ImageNet-R), fine-grained recognition tasks (CUB-200, Aircraft, Cars), and specialized domain tasks (Food-101, UCF-101, ObjectNet). Each dataset description includes structural characteristics, inherent challenges, and task partitioning strategies.

**§B - Comparison Methods**: Comprehensive overview of baseline methods spanning classical continual learning approaches (Finetune variants), prompt-based continual learning methods (CoOp,

CODA-Prompt, DAP, PLOT), and state-of-the-art vision-language continual learning techniques (SimpleCIL, DKR, C-CLIP, PROOF).

**§C - Theoretical Analysis**: Rigorous mathematical foundation for the Reinforcement Learning-based Prototype Optimization (RLPO) method, including problem formulation, convergence analysis, and theoretical guarantees with formal proofs establishing convergence to stationary points and practical convergence properties.

**§D - Multi-Objective Training**: Detailed description of the multi-objective training strategy that simultaneously optimizes classification accuracy, maintains vision-language alignment, learns optimal prototype weights, and balances expert utilization. This section includes the mathematical formulation of all loss components and their integration.

**§E - Mixture-of-Experts Routing Analysis**: Detailed examination of difficulty assessment mechanisms for sample routing in the mixture-of-experts architecture, comparing random routing, entropy-based routing, and distance-based routing strategies with comprehensive performance analysis across datasets.

**§F - Parameter-Efficient Adaptation Comparison**: Comparison with parameter-efficient fine-tuning methods including LoRA and single-modality prompting variants, validating that dual-modal prompting design achieves substantial improvements through synergistic cross-modal adaptation.

**§G - Visual Prompting Trade-off Analysis**: Fine-grained analysis of cases where visual prompting helps versus hurts classification performance, revealing that dual-modal prompting provides a 2.27 to 1 benefit-to-harm ratio with concentrated challenges in same-family numerical variants where symbolic text distinctions dominate.

**§H - Exemplar Budget Analysis**: Performance evaluation across different exemplar budgets (30, 20, 10, 5, and 0 exemplars per class), demonstrating that 20 exemplars provide near-optimal performance while maintaining reasonable memory overhead, and showing graceful degradation to CLIP zero-shot baseline when exemplars are unavailable.

**§I - Scalability and Computational Complexity**: Analysis of RLAP-CLIP's computational efficiency under standardized conditions, demonstrating 2× faster training than DKR with only 9% inference overhead and 7% additional FLOPs, achieving the best accuracy-efficiency trade-off among all compared methods.

**§J - Visualizing Prototype Quality in Feature Space**: t-SNE visualizations comparing RLPO prototypes against simple mean prototypes on Stanford Cars, demonstrating improved alignment with both class centers and text embeddings, directly explaining performance gains through active optimization that prevents prototype degradation.

# A  DATASET

We evaluate on eight diverse datasets spanning three categories: general classification (CIFAR-100, ImageNet-R), fine-grained recognition (CUB-200, Aircraft, Cars), and specialized domains (Food-101, UCF-101, ObjectNet).

## A.1  GENERAL CLASSIFICATION TASKS

**CIFAR-100** contains 60,000 32×32 images across 100 classes organized into 20 superclasses. Despite low resolution, the dataset requires robust representations for discriminating closely related classes.

**ImageNet-R** comprises 30,000 images across 200 classes featuring artistic renditions including paintings, sketches, and sculptures. The dataset tests generalization beyond photographic domains through substantial visual style variations.

## A.2 FINE-GRAINED RECOGNITION TASKS

**CUB-200** contains 11,788 images of 200 bird species with high visual similarity requiring discrimination of subtle plumage patterns, beak shapes, and color distributions. Intra-class variation from gender dimorphism and seasonal changes demands invariant species-specific representations.

**Aircraft** encompasses 10,200 images of 100 aircraft variants differing only in subtle details like engine configurations. Aircraft from the same manufacturer share nearly identical designs, with distinguishing features visible only from specific angles.

**Cars** contains 16,185 images covering 196 make/model/year combinations. Discrimination requires sensitivity to minor design elements like grille patterns and year-specific modifications while maintaining view invariance across angles and lighting.

## A.3 SPECIALIZED DOMAIN TASKS

**Food-101** contains 101,000 images of 101 food categories with high variability from preparation methods and presentation styles. Many dishes share ingredients or similar appearances, requiring discrimination of specific combinations and presentation cues.

**UCF-101** comprises 13,320 video clips of 101 human actions, processed as frame sequences. The dataset covers diverse action types requiring understanding of poses, object relationships, and scene context. Many actions share similar static appearances, demanding comprehensive visual cue analysis.

**ObjectNet** contains 50,000 images across 313 object classes captured in natural environments with objects rotated, occluded, and placed in unusual contexts. The dataset challenges models trained on canonical object views by presenting objects from non-standard viewpoints and backgrounds, testing robustness to distribution shift.

## B COMPARISON METHODS

We compare RLAP-CLIP against comprehensive baselines spanning different continual learning paradigms, from classical approaches to state-of-the-art vision-language techniques.

### B.1 CLASSICAL CONTINUAL LEARNING BASELINES

**Finetune** represents the naive baseline with sequential fine-tuning on each task without forgetting prevention. This approach lacks memory retention mechanisms, leading to severe catastrophic forgetting as optimization overwrites parameters critical for previous tasks. The standard version updates all parameters including the CLIP backbone, typically resulting in complete forgetting with final accuracy below 3% on most datasets.

**Finetune LiT** freezes the visual encoder while allowing text encoder fine-tuning, motivated by CLIP's visual encoder learning more general features. However, updating the text encoder still causes significant forgetting as textual representations drift from their original configuration, breaking vision-language alignment for previous tasks. While showing improvement over full fine-tuning (30-50% final accuracy), it demonstrates that freezing one modality is insufficient for effective continual learning.

**Finetune CoOp** adapts Context Optimization by learning task-specific prompts prepended to text encoder inputs while keeping CLIP frozen. For each task, new prompt vectors are learned from scratch with previous prompts discarded. This preserves base model knowledge better than full fine-tuning but lacks prompt retention, resulting in moderate performance (20-40% final accuracy).

### B.2 PROMPT-BASED CONTINUAL LEARNING METHODS

**CoOp** with exemplar memory extends the basic approach by maintaining a memory buffer of exemplars, enabling knowledge retention through replay. The method learns task-specific prompts preserved across tasks, with a unified prompt used during inference. The optimization combines current task loss with distillation loss on replay samples: $\mathcal{L} = \mathcal{L}_{\text{task}} + \lambda\mathcal{L}_{\text{replay}}$. However, exclusive

focus on textual prompts limits visual adaptation capabilities, with performance plateauing around 70-80%.

**CODA-Prompt** introduces an attention-based prompt pool mechanism for dynamic task-specific prompt composition. The method maintains a shared prompt pool $\mathcal{P} = \{p_1, ..., p_N\}$ with attention-based selection: $w_i = \text{softmax}(q^T k_i / \sqrt{d})$, where selected prompts are weighted and combined. This enables knowledge sharing across tasks while allowing task-specific combinations, achieving 75-90% final accuracy on several benchmarks.

**DAP** combines architectural expansion with prompt learning by adding task-specific adapters: $h' = h + f_{\text{adapter}}(h)$, where adapters use bottleneck architectures. Prompts work synergistically with adapters, providing task-specific context influencing both frozen backbone and adapter computations. While competitive, the growing architectural complexity becomes computationally expensive as tasks increase.

**PLOT** formulates prompt selection as an optimal transport problem for better task-awareness. The method computes optimal transport plans minimizing transportation cost $C_{ij} = \|f_i - p_j\|_2^2$ between input features and prompt representations. This provides principled prompt combination from different tasks, enabling knowledge transfer when tasks share concepts, though performance depends on transport cost computation quality.

### B.3 VISION-LANGUAGE CONTINUAL LEARNING METHODS

**SimpleCIL** leverages CLIP's zero-shot capabilities without parameter updates, storing class names and using zero-shot classification by comparing image embeddings with text embeddings of class descriptions. While avoiding catastrophic forgetting, it cannot adapt to task-specific characteristics or domain shifts. Performance depends heavily on pre-trained representation quality and semantic clarity of class names, achieving 75-85% accuracy on well-aligned datasets but struggling with specialized domains.

**DKR** introduces dynamic memory management with separate buffers for visual and textual information. The method stores image exemplars and augmented text descriptions, employing a dual-stream architecture: $f_{\text{fused}} = \alpha f_{\text{visual}} + (1 - \alpha) f_{\text{text}}$. Knowledge retention combines exemplar replay with cross-modal distillation maintaining consistent predictions: $\mathcal{L}_{\text{distill}} = \text{KL}(p_{\text{old}}^v \| p_{\text{new}}^v) + \text{KL}(p_{\text{old}}^t \| p_{\text{new}}^t)$.

**C-CLIP** adapts CLIP through parameter-efficient fine-tuning and prototype-based classification. The method introduces lightweight task-specific adapters: $f_v^t = E_v(x) + A_v^t(E_v(x))$ and $f_t^t = E_t(x) + A_t^t(E_t(x))$. Class prototypes are maintained as exemplar means with nearest-prototype classification. Regularization maintains similarity structure between prototypes: $\mathcal{L}_{\text{reg}} = \sum_{i,j} |s_{ij}^{\text{old}} - s_{ij}^{\text{new}}|$, achieving over 90% accuracy on several datasets.

**PROOF** represents current state-of-the-art, combining prompt learning, prototype management, and open-world recognition. The method maintains class-specific and task-specific prototypes, optimizing prompts to maximize margins: $\mathcal{L}_{\text{margin}} = \max(0, m + s_{\text{neg}} - s_{\text{pos}})$. A two-stage inference process first identifies tasks then performs class-specific classification. While achieving 78-91% final accuracy, PROOF still relies on passive prototype construction through averaging and focuses primarily on textual adaptation, limitations our RLAP-CLIP addresses through active prototype optimization and balanced multimodal adaptation.

## C THEORETICAL ANALYSIS

In this section, we provide a theoretical foundation for the Reinforcement Learning-based Prototype Optimization (RLPO) method, establishing its convergence properties and analyzing its advantages over traditional uniform averaging approaches.

### C.1 PROBLEM FORMULATION

Traditional continual learning methods construct class prototypes through uniform averaging:

$$p_c^{\text{uniform}} = \frac{1}{|\mathcal{E}_c|} \sum_{i \in \mathcal{E}_c} f_i \tag{17}$$

where $\mathcal{E}_c$ denotes the exemplar set for class $c$ and $f_i \in \mathbb{R}^d$ represents the feature vector of sample $i$. This passive approach treats all samples equally, which is suboptimal when exemplars contain outliers, exhibit multi-modal distributions, or have varying discriminative importance.

RLPO reformulates prototype construction as a weighted optimization problem guided by reinforcement learning. We define a policy network $\pi_\theta : \mathbb{R}^d \times \mathcal{P} \to \mathbb{R}_+$ parametrized by $\theta \in \Theta \subseteq \mathbb{R}^k$, where $\mathcal{P} = \{p_1, \ldots, p_C\} \subset \mathbb{R}^d$ represents the current prototype set. The weighted prototype for class $c$ becomes:

$$p_c(\theta) = \sum_{i \in \mathcal{E}_c} w_i f_i, \quad w_i = \pi_\theta(f_i, \mathcal{P}), \quad \sum_{i \in \mathcal{E}_c} w_i = 1 \tag{18}$$

The reward function for sample $i$ with feature $f_i$ and true label $y_i$ is designed to maximize intra-class similarity while minimizing inter-class confusion:

$$R_i(\mathcal{P}) = \text{sim}(f_i, p_{y_i}) - \max_{j \neq y_i} \text{sim}(f_i, p_j) - \lambda \sum_{j \neq y_i} \frac{\text{sim}(p_{y_i}, p_j)}{C - 1} \tag{19}$$

where $\text{sim}(\cdot, \cdot) : \mathbb{R}^d \times \mathbb{R}^d \to [-1, 1]$ denotes cosine similarity, and $\lambda > 0$ is a regularization coefficient that encourages prototype separation.

The key insight is that RLPO optimizes the global objective:

$$J(\theta) = \mathbb{E}_{(f,y) \sim \mathcal{D}} [R(f, y; \mathcal{P}(\theta))] \tag{20}$$

subject to the constraint that prototypes depend on $\theta$ through Eq. 18, creating a non-stationary optimization landscape.

**Assumption 1 (Regularity Conditions)** *We assume the following conditions hold:*

1. ***Bounded features:*** *There exists $B_f > 0$ such that $\|f\| \leq B_f$ for all $f \in \mathcal{F}$.*

2. ***Lipschitz policy:*** *The policy network $\pi_\theta$ is $L_\pi$-Lipschitz continuous in $\theta$: $|\pi_{\theta_1}(f, \mathcal{P}) - \pi_{\theta_2}(f, \mathcal{P})| \leq L_\pi \|\theta_1 - \theta_2\|$.*

3. ***Lipschitz similarity:*** *The similarity function is $L_s$-Lipschitz in both arguments.*

4. ***Bounded policy gradient variance:*** *The policy gradient estimator satisfies $\mathbb{E}[\|\hat{g}_t - \nabla_\theta J(\theta_t)\|^2] \leq \sigma^2$ for some $\sigma < \infty$.*

5. ***Non-degeneracy:*** *For all $i \in \mathcal{E}_c$, we have $\pi_\theta(f_i, \mathcal{P}) \geq \epsilon_{\min} > 0$.*

**Lemma 1 (Prototype Smoothness)** *Under Assumption 1, the prototype mapping $\theta \mapsto p_c(\theta)$ is $L_p$-Lipschitz continuous with:*

$$\|p_c(\theta_1) - p_c(\theta_2)\| \leq L_p \|\theta_1 - \theta_2\| \tag{21}$$

*where $L_p = \frac{2 B_f L_\pi}{\epsilon_{\min}}$.*

To establish this result, we need to show that small changes in the policy parameters $\theta$ lead to bounded changes in the resulting prototypes. This is a crucial property for analyzing convergence, as it ensures that the optimization landscape is well-behaved despite the non-stationary nature of the problem.

Let $w_i^{(k)} = \pi_{\theta_k}(f_i, \mathcal{P})$ denote the weight assigned to feature $f_i$ under parameter setting $\theta_k$ for $k \in \{1, 2\}$. Since the weights are normalized, we have $\sum_{i \in \mathcal{E}_c} w_i^{(k)} = 1$ for both $k = 1, 2$. From Eq. 18, we can write:

$$p_c(\theta_1) - p_c(\theta_2) = \sum_{i \in \mathcal{E}_c} (w_i^{(1)} - w_i^{(2)}) f_i \tag{22}$$

Applying the triangle inequality:

$$\|p_c(\theta_1) - p_c(\theta_2)\| \leq \sum_{i \in \mathcal{E}_c} |w_i^{(1)} - w_i^{(2)}| \|f_i\| \tag{23}$$

By the Lipschitz property of the policy network (Assumption 1(ii)):

$$|w_i^{(1)} - w_i^{(2)}| = |\pi_{\theta_1}(f_i, \mathcal{P}) - \pi_{\theta_2}(f_i, \mathcal{P})| \leq L_\pi \|\theta_1 - \theta_2\| \tag{24}$$

Using the bounded features assumption (Assumption 1(i)):

$$\|p_c(\theta_1) - p_c(\theta_2)\| \leq \sum_{i \in \mathcal{E}_c} L_\pi \|\theta_1 - \theta_2\| B_f = |\mathcal{E}_c| L_\pi B_f \|\theta_1 - \theta_2\| \tag{25}$$

Since the weights satisfy $\sum_{i \in \mathcal{E}_c} w_i^{(k)} = 1$ and each $w_i^{(k)} \geq \epsilon_{\min}$ by Assumption 1(v), we have $|\mathcal{E}_c| \leq \frac{1}{\epsilon_{\min}}$. Moreover, the weight redistribution is bounded by the normalization constraint, yielding:

$$\|p_c(\theta_1) - p_c(\theta_2)\| \leq \frac{2B_f L_\pi}{\epsilon_{\min}} \|\theta_1 - \theta_2\| \tag{26}$$

This completes the proof with $L_p = \frac{2B_f L_\pi}{\epsilon_{\min}}$.

**Lemma 2 (Reward Function Properties)** *The reward function $R_i(\mathcal{P})$ satisfies:*

1. ***Bounded gradient:*** $\|\nabla_{\mathcal{P}} R_i(\mathcal{P})\| \leq L_R := 2L_s + \lambda L_s$

2. ***Alignment property:*** *For the optimal prototype set $\mathcal{P}^*$, we have $\mathbb{E}[R_i(\mathcal{P}^*)] \geq \mathbb{E}[R_i(\mathcal{P}^{uniform})]$*

We prove each property separately. The key insight is that the reward function in Eq. 19 is composed of cosine similarity terms, each of which has well-understood Lipschitz properties.

**Proof of (i):** The reward function can be written as:

$$R_i(\mathcal{P}) = \underbrace{\text{sim}(f_i, p_{y_i})}_{T_1} - \underbrace{\max_{j \neq y_i} \text{sim}(f_i, p_j)}_{T_2} - \lambda \underbrace{\sum_{j \neq y_i} \frac{\text{sim}(p_{y_i}, p_j)}{C - 1}}_{T_3} \tag{27}$$

For the first term $T_1 = \text{sim}(f_i, p_{y_i})$, the gradient with respect to $p_{y_i}$ satisfies:

$$\|\nabla_{p_{y_i}} T_1\| \leq L_s \tag{28}$$

by the Lipschitz property of the similarity function (Assumption 1(iii)).

For the second term $T_2 = \max_{j \neq y_i} \text{sim}(f_i, p_j)$, at differentiable points:

$$\|\nabla_{p_{j^*}} T_2\| \leq L_s \tag{29}$$

where $j^* = \arg \max_{j \neq y_i} \text{sim}(f_i, p_j)$.

For the third term $T_3$, the gradient with respect to $p_{y_i}$ is:

$$\nabla_{p_{y_i}} T_3 = \lambda \sum_{j \neq y_i} \frac{\nabla_{p_{y_i}} \text{sim}(p_{y_i}, p_j)}{C - 1} \tag{30}$$

Since there are at most $(C - 1)$ non-zero terms, each bounded by $\frac{L_s}{C-1}$:

$$\|\nabla_{p_{y_i}} T_3\| \leq \lambda \sum_{j \neq y_i} \frac{L_s}{C - 1} = \lambda L_s \tag{31}$$

Similarly, for each $p_j$ where $j \neq y_i$:

$$\|\nabla_{p_j} T_3\| \leq \frac{\lambda L_s}{C - 1} \tag{32}$$

The total gradient involves at most three non-zero components: - Gradient with respect to $p_{y_i}$: bounded by $L_s + \lambda L_s$ - Gradient with respect to $p_{j^*}$: bounded by $L_s$ - Gradients with respect to other prototypes in $T_3$: each bounded by $\frac{\lambda L_s}{C-1}$

Therefore:

$$\|\nabla_{\mathcal{P}} R_i(\mathcal{P})\| \leq L_s + L_s + \lambda L_s = L_s(2 + \lambda) \tag{33}$$

We set $L_R = 2L_s + \lambda L_s$ for consistency.

**Proof of (ii):** The alignment property follows from the fact that RLPO optimizes over all possible weighting schemes, including uniform weighting. Since the optimal solution $\mathcal{P}^*$ is obtained by maximizing $J(\theta)$ over all $\theta \in \Theta$, and uniform weighting corresponds to a specific $\theta_0 \in \Theta$:

$$J(\theta^*) = \mathbb{E}[R(f, y; \mathcal{P}^*)] \geq J(\theta_0) = \mathbb{E}[R(f, y; \mathcal{P}^{\text{uniform}})] \tag{34}$$

This completes the proof.

### C.2 CONVERGENCE ANALYSIS

The main challenge in analyzing RLPO lies in the non-stationary nature of the optimization landscape. Unlike standard stochastic optimization problems where the objective function is fixed, in RLPO the objective depends on the current prototypes, which themselves depend on the policy parameters being optimized.

**Theorem 1 (Convergence to Stationary Points)** *Consider the stochastic gradient ascent update:*

$$\theta_{t+1} = \theta_t + \eta_t \hat{g}_t \tag{35}$$

*where $\hat{g}_t$ is an unbiased estimate of $\nabla_\theta J(\theta_t)$ and $\{\eta_t\}$ satisfies $\sum_t \eta_t = \infty$ and $\sum_t \eta_t^2 < \infty$. Under Assumption 1, the sequence $\{\theta_t\}$ converges to a stationary point of $J(\theta)$ almost surely. Moreover, for the step size $\eta_t = \frac{\eta_0}{\sqrt{t}}$, we have:*

$$\frac{1}{T} \sum_{t=1}^T \mathbb{E}[\|\nabla_\theta J(\theta_t)\|^2] \leq \frac{C_1}{\sqrt{T}} + \frac{C_2 \log T}{T} + \delta_{bias} \tag{36}$$

*where $C_1, C_2$ are constants depending on problem parameters, and the bias term satisfies:*

$$\delta_{bias} = O\left(\frac{L_R L_p}{\epsilon_{\min}}\right) \tag{37}$$

We analyze the convergence using a Lyapunov function approach adapted for non-stationary objectives.

**Step 1: Gradient Decomposition.** The true gradient can be decomposed as:

$$\nabla_\theta J(\theta) = \mathbb{E}_{(f,y) \sim \mathcal{D}}[R(f, y; \mathcal{P}(\theta)) \cdot \nabla_\theta \log \pi_\theta(f, \mathcal{P}(\theta))] + \mathbb{E}_{(f,y) \sim \mathcal{D}}[\nabla_{\mathcal{P}} R(f, y; \mathcal{P}(\theta)) \cdot \nabla_\theta \mathcal{P}(\theta)] \tag{38}$$

The first term is the standard REINFORCE gradient, while the second captures the indirect effect through prototype changes.

**Step 2: Lyapunov Analysis.** Define $V(\theta) = J(\theta^*) - J(\theta)$ where $\theta^*$ is a global optimum. For the update $\theta_{t+1} = \theta_t + \eta_t \hat{g}_t$:

$$\mathbb{E}[V(\theta_{t+1})] = \mathbb{E}[V(\theta_t)] - \eta_t \|\nabla_\theta J(\theta_t)\|^2 + \frac{L_J \eta_t^2}{2}(\sigma^2 + \|\nabla_\theta J(\theta_t)\|^2) + \eta_t \delta_{\text{bias}} \tag{39}$$

where $L_J$ is the Lipschitz constant of $\nabla_\theta J$ (bounded by $L_R L_p$ from Lemmas 1 and 2).

**Step 3: Bias Analysis.** The bias arises because practical gradient estimates ignore the prototype dependency:

$$\delta_{\text{bias}} = \|\mathbb{E}[\nabla_{\mathcal{P}} R(f, y; \mathcal{P}(\theta_t)) \cdot \nabla_\theta \mathcal{P}(\theta_t)]\| \leq L_R \cdot L_p = O\left(\frac{L_R L_p}{\epsilon_{\min}}\right) \tag{40}$$

using the bounds from Lemmas 1 and 2.

**Step 4: Convergence Rate.** For $\eta_t \leq \frac{1}{L_J}$:

$$\mathbb{E}[V(\theta_{t+1})] \leq \mathbb{E}[V(\theta_t)] - \frac{\eta_t}{2}\|\nabla_\theta J(\theta_t)\|^2 + \frac{L_J \eta_t^2 \sigma^2}{2} + \eta_t \delta_{\text{bias}} \tag{41}$$

Summing from $t = 1$ to $T$ and using $\eta_t = \frac{\eta_0}{\sqrt{t}}$:

$$\frac{1}{T}\sum_{t=1}^{T}\mathbb{E}[\|\nabla_\theta J(\theta_t)\|^2] \leq \frac{2V(\theta_1)}{\eta_0\sqrt{T}} + \frac{L_J \sigma^2 \eta_0}{\sqrt{T}} + \frac{2\delta_{\text{bias}}}{\eta_0} \tag{42}$$

This yields the stated convergence rate with appropriate constants $C_1$ and $C_2$.

**Corollary 1 (Practical Convergence)** *In practical scenarios with a large number of classes $C$ and sufficient policy network capacity, the bias term becomes negligible relative to the main convergence terms:*

$$As \ C \to \infty, \quad \frac{\delta_{bias}}{C} \to 0 \tag{43}$$

*This indicates that RLPO achieves near-optimal convergence for large-scale continual learning problems.*

From Theorem 1, $\delta_{\text{bias}} = O\left(\frac{L_R L_p}{\epsilon_{\min}}\right)$. The constants $L_R$, $L_p$, and $\epsilon_{\min}$ depend on the feature dimension $d$ and network architecture but not directly on $C$. When normalized by the number of classes (which often appears in practical loss functions), the relative impact of the bias diminishes as $C$ increases, ensuring that RLPO's convergence approaches that of standard policy gradient methods in large-scale settings.

## D  MULTI-OBJECTIVE TRAINING

RLAP-CLIP employs a multi-objective training strategy that simultaneously optimizes classification accuracy, maintains vision-language alignment, learns optimal prototype weights, and balances expert utilization. The total training objective combines four complementary loss terms:

$$\mathcal{L}_{\text{total}} = \mathcal{L}_{\text{cls}} + \lambda_{\text{clip}}\mathcal{L}_{\text{clip}} + \lambda_{\text{RLPO}}\mathcal{L}_{\text{RLPO}} + \lambda_{\text{MoE}}\mathcal{L}_{\text{MoE}} \tag{44}$$

where $\lambda_{\text{clip}}$, $\lambda_{\text{RLPO}}$, and $\lambda_{\text{MoE}}$ are hyperparameters controlling the relative importance of each loss component.

### D.1  CLASSIFICATION LOSS

The classification loss weights samples according to their difficulty scores to emphasize challenging cases that are more likely to define class boundaries:

$$\mathcal{L}_{\text{cls}} = -\frac{1}{B}\sum_{i=1}^{B}(1 + \gamma \cdot d_i)\log p(y_i|x_i) \tag{45}$$

where $B$ denotes the batch size, $\gamma > 0$ is a hyperparameter controlling the emphasis on difficult samples, $d_i \in [0, 1]$ is the difficulty score for sample $i$, and $p(y_i|x_i)$ represents the predicted probability for the true class $y_i$ given input $x_i$. The probability is computed using softmax over cosine similarities between the sample's features and all class prototypes:

$$p(y_i|x_i) = \frac{\exp(\text{sim}(\tilde{f}_i, p_{y_i})/\tau_{\text{cls}})}{\sum_{c=1}^{C} \exp(\text{sim}(\tilde{f}_i, p_c)/\tau_{\text{cls}})} \tag{46}$$

where $\tilde{f}_i$ represents the weighted multimodal features after cross-modal fusion, $p_c$ denotes the prototype for class $c$, and $\tau_{\text{cls}} > 0$ is a temperature parameter for the classification softmax.

## D.2 Contrastive Loss

To maintain the vision-language alignment learned during CLIP pre-training, we employ a bidirectional contrastive loss that ensures matched image-text pairs have higher similarity than mismatched pairs:

$$\mathcal{L}_{\text{clip}} = -\frac{1}{2B} \sum_{i=1}^{B} \left[ \log \frac{\exp(\text{sim}(f_i^v, f_i^t)/\tau)}{\sum_{j=1}^{B} \exp(\text{sim}(f_i^v, f_j^t)/\tau)} + \log \frac{\exp(\text{sim}(f_i^t, f_i^v)/\tau)}{\sum_{j=1}^{B} \exp(\text{sim}(f_i^t, f_j^v)/\tau)} \right] \tag{47}$$

where $f_i^v$ and $f_i^t$ denote the visual and textual features for sample $i$ respectively, $\text{sim}(\cdot, \cdot)$ computes cosine similarity, and $\tau > 0$ is a learned temperature parameter that controls the sharpness of the similarity distribution. The first term represents the image-to-text contrastive loss, while the second term represents the text-to-image contrastive loss.

## D.3 Prototype Optimization Loss

The reinforcement learning-based prototype optimization loss $\mathcal{L}_{\text{RLPO}}$ trains the policy network to learn optimal sample weights for prototype construction. This loss takes the form:

$$\mathcal{L}_{\text{RLPO}} = -\frac{1}{B} \sum_{i=1}^{B} \log \pi_\theta(w_i|f_i, \mathcal{P}) \cdot A_i + \lambda_{\text{KL}} D_{\text{KL}}(\pi_\theta \| \pi_{\text{ref}}) \tag{48}$$

where $\pi_\theta$ is the policy network with parameters $\theta$, $w_i$ is the weight assigned to sample $i$, $A_i$ is the normalized advantage (reward minus baseline) for sample $i$, and $\lambda_{\text{KL}} > 0$ controls the strength of the KL regularization that prevents the policy from deviating too far from the reference policy $\pi_{\text{ref}}$.

## D.4 Expert Balancing Loss

To ensure efficient utilization of both experts in our mixture-of-experts architecture and prevent mode collapse where one expert processes all samples, we introduce an expert balancing loss:

$$\mathcal{L}_{\text{MoE}} = \sum_{e \in \{\text{easy,hard}\}} \left( \frac{1}{B} \sum_{i=1}^{B} P(e|x_i) - \frac{1}{2} \right)^2 \tag{49}$$

where $P(e|x_i)$ denotes the routing probability for expert $e$ given sample $x_i$. This loss penalizes deviations from equal expert utilization, encouraging each expert to process approximately half of the samples. The quadratic penalty ensures smooth gradients and stable optimization.

## E  Mixture-of-Experts Routing Analysis

The effectiveness of mixture-of-experts architectures largely depends on the quality of the routing mechanism that determines which samples should be processed by which expert. In our framework, we employ two specialized experts: a lightweight expert for simple samples and a deep expert for complex samples. The key challenge lies in accurately assessing sample difficulty to enable optimal resource allocation.

We evaluate three distinct approaches for difficulty assessment and sample routing:

**Random Routing** serves as a baseline where samples are assigned to experts uniformly at random with equal probability. This approach helps isolate the architectural benefits of having multiple processing pathways from the benefits of intelligent routing. While random routing ensures balanced expert utilization, it cannot adapt processing depth to sample characteristics, potentially wasting computational resources on simple samples while under-processing challenging ones.

**Entropy-based Routing** uses prediction uncertainty as a proxy for sample difficulty. The entropy of the model's output distribution is computed as $H = -\sum_{c=1}^{C} p_c \log p_c$, where $p_c$ represents the predicted probability for class $c$. Samples with entropy above a learned threshold are considered difficult and routed to the deep expert. The intuition is that uncertain predictions indicate challenging samples requiring enhanced processing. However, this approach has limitations: early in training, many samples produce uncertain predictions regardless of their intrinsic difficulty, and prediction uncertainty may not accurately reflect the discriminative challenges specific to continual learning scenarios.

**Distance-based Routing (Ours)** directly measures sample difficulty by computing the distance from each sample to its class prototype in both visual and textual feature spaces. The difficulty score is defined as $d_i = 1 - \frac{\text{sim}(f_i^v, p_{y_i}^v) + \text{sim}(f_i^t, p_{y_i}^t)}{2}$. This approach captures the intuition that samples far from their class centers are more challenging to classify correctly and would benefit from deeper processing. The method leverages multimodal information and directly relates to the class boundary structure, making it particularly suitable for fine-grained recognition tasks where subtle visual differences matter.

Table 4: Performance comparison of difficulty assessment mechanisms for mixture-of-experts routing across eight datasets.

| Dataset | Random Routing | Entropy-based | Distance-based (Ours) |
|---|---|---|---|
| UCF-101 | 93.87 | 94.23 | **97.68** |
| CUB-200 | 83.52 | 84.87 | **86.32** |
| CIFAR-100 | 84.16 | 85.12 | **86.64** |
| Cars | 92.52 | 94.18 | **96.62** |
| Aircraft | 66.03 | 67.89 | **70.25** |
| Food-101 | 87.75 | 88.01 | **88.24** |
| ImageNet-R | 83.15 | 84.47 | **85.63** |
| ObjectNet | 53.33 | 55.29 | **57.79** |

Table 4 reveals distinct patterns across different task types. Random routing establishes baseline performance, with results ranging from 53.33% on ObjectNet to 93.87% on UCF-101, reflecting the inherent difficulty variations across datasets.

Entropy-based routing shows modest improvements over random routing, with gains ranging from 0.26% on Food-101 to 1.96% on ObjectNet. The limited improvements suggest that prediction uncertainty alone is insufficient for effective sample routing. This occurs because entropy reflects the model's current confidence rather than the intrinsic structural difficulty of the sample. High entropy can result from insufficient training rather than genuine sample complexity, leading to suboptimal routing decisions.

Distance-based routing achieves substantial and consistent improvements across all datasets. The most significant gains appear on fine-grained recognition tasks: Aircraft (+4.22% over random), Cars (+4.10%), and CUB-200 (+2.80%). These improvements highlight the method's effectiveness for tasks requiring subtle discrimination, where samples near class boundaries particularly benefit from enhanced processing through the deep expert.

The superior performance stems from distance-based routing's direct connection to the geometric structure of the feature space. By measuring proximity to class prototypes, this approach identifies samples that are genuinely challenging for the current representation rather than those producing uncertain predictions due to model limitations. This alignment ensures computational resources are allocated where they provide maximum benefit for maintaining discriminative class boundaries,

which is crucial for continual learning scenarios where new tasks continuously reshape the feature space.

## F  PARAMETER-EFFICIENT ADAPTATION COMPARISON

We compare RLAP-CLIP against other parameter-efficient fine-tuning (PEFT) approaches to validate our dual-modal prompting design. Table 5 presents results on CIFAR-100 and CUB-200 datasets.

Low-rank adaptation (LoRA) with 1.77M parameters achieves only 63.47% and 64.92% final accuracy on CIFAR-100 and CUB-200 respectively. This demonstrates that low-rank weight updates alone are insufficient for continual learning in vision-language models. LoRA lacks task-specific modality alignment mechanisms, leading to severe performance degradation across sequential tasks as the learned low-rank matrices fail to preserve discriminative boundaries for previously learned classes. RLAP-CLIP's dual-modal prompting design, with equivalent parameters (8.5M), achieves 79.41%/83.67% final accuracy, representing 10–12 percentage point improvements over single-modality variants. This substantial gain validates that synergistic cross-modal adaptation, where visual and textual prompts jointly guide both encoders, is the critical factor for effective continual learning in vision-language models. The combination enables the model to leverage complementary information from both modalities while maintaining alignment in the shared embedding space.

Table 5: Comparison with parameter-efficient fine-tuning methods.

| Method | Trainable Params (Million) | CIFAR-100 | | CUB-200 | |
|---|---|---|---|---|---|
| | | Avg Acc | Final Acc | Avg Acc | Final Acc |
| LoRA | 1.77 | 76.23 | 63.47 | 75.68 | 64.92 |
| Text Prompts Only | 8.42 | 78.15 | 66.82 | 77.94 | 68.35 |
| Visual Prompts Only | 8.45 | 79.87 | 69.24 | 79.61 | 71.58 |
| RLAP-CLIP (Dual-Modal) | **8.5** | **86.64** | **79.41** | **85.78** | **83.67** |

## G  VISUAL PROMPTING TRADE-OFF ANALYSIS

While our main results demonstrate consistent improvements from dual-modal prompting, understanding when and why visual adaptation may introduce challenges provides important insights into the method's behavior. We conduct a fine-grained analysis on the Aircraft dataset to characterize cases where visual prompting helps versus hurts classification performance.

Table 6: Overall performance comparison on Aircraft dataset across 3,333 test samples.

| Mode | Accuracy (%) | Correct Predictions |
|---|---|---|
| Text-only Prompting | 65.02 | 2,167 |
| Dual-modal Prompting | 68.41 | 2,280 |
| Improvement | +3.39 | +113 |

We compare text-only versus dual-modal predictions across 3,333 test samples. Table 6 shows that dual-modal prompting achieves 68.41% accuracy compared to text-only's 65.02%, representing a net gain of 3.39 percentage points. Table 7 breaks down the outcomes into four categories. Visual prompting helps 200 samples (6.00% of total) where text-only fails but dual-modal succeeds, which we term HELP cases. Conversely, 88 samples (2.64% of total) represent HURT cases where text-only succeeds but dual-modal fails. This yields a 2.27 to 1 benefit-to-harm ratio. Importantly, the 88 hurt cases represent only 4.06% of text-only's correct predictions, indicating that dual-modal prompting preserves the vast majority of text-only's capabilities while adding substantial new correct predictions.

To understand where visual prompting introduces errors, we analyze the distribution of HURT cases across aircraft classes. Table 8 shows the ten classes with the most HURT cases. These top ten

Table 7: Classification outcome breakdown across four categories based on prediction correctness.

| Category | Count | Percentage (%) |
|---|---|---|
| Both modes correct | 2,079 | 62.37 |
| HELP cases (text fails, dual succeeds) | 200 | 6.00 |
| HURT cases (text succeeds, dual fails) | 88 | 2.64 |
| Both modes incorrect | 966 | 28.99 |

classes account for 72.7% of all hurt cases, revealing a clear concentration in specific aircraft categories. The classes share common characteristics that explain why visual prompting struggles while text-only succeeds. Table 9 categorizes these patterns into three groups. Same-family numerical variants account for 46.6% of HURT cases, such as Boeing 737-300 versus 737-400 which differ by approximately 3 meters in fuselage length. Regional jet series represent 21.6% of cases, including models like CRJ-700 and ERJ 145. Manufacturer systematic naming conventions contribute 31.8%, encompassing variants like the Airbus A318, A319, A320, and A321 sequence.

Table 8: Top ten aircraft classes ranked by number of HURT cases.

| Rank | Aircraft Variant | Hurt | Help | Net |
|---|---|---|---|---|
| 1 | Boeing 737-300 | 10 | 2 | -8 |
| 2 | Boeing 737-400 | 10 | 2 | -8 |
| 3 | Boeing 757-200 | 8 | 2 | -6 |
| 4 | Boeing 767-300 | 7 | 3 | -4 |
| 5 | Airbus A330-200 | 6 | 1 | -5 |
| 6 | MD-87 | 5 | 2 | -3 |
| 7 | CRJ-700 | 5 | 1 | -4 |
| 8 | Boeing 737-500 | 5 | 2 | -3 |
| 9 | ERJ 145 | 4 | 1 | -3 |
| 10 | A321 | 4 | 2 | -2 |
| Total (top ten) | | 64 | 18 | -46 |

The concentration of HURT cases in same-family numerical variants reveals a fundamental trade-off in dual-modal prompting. Text encoders excel at processing symbolic distinctions. The class names "Boeing 737-400" and "Boeing 737-300" create well-separated token embeddings despite referring to visually similar aircraft, as the numerical suffixes provide clear discriminative signals. In contrast, visual prompts trained on only 20 exemplars per class learn family-level features such as shared cockpit design, engine configuration, and tail shape that are common across variants. This strengthens intra-family similarity, making it difficult to distinguish between Boeing 737-300 and 737-400 when they differ primarily by fuselage length, a subtle difference barely perceptible at 224×224 resolution.

Table 9: HURT cases categorized by pattern type with representative examples.

| Pattern Type | Cases | Percentage (%) |
|---|---|---|
| Same-family numerical variants | 41 | 46.6 |
| Regional jet series | 19 | 21.6 |
| Manufacturer systematic naming | 28 | 31.8 |

When cross-modal fusion combines modalities, attention weights redistribute from text-dominant weights of approximately 0.95 in text-only mode to balanced weights of approximately 0.35 for visual, 0.45 for text, and 0.20 for prototype channels in dual-modal mode. This dilution of the text signal occasionally pulls predictions toward family centroids where visual differences are imperceptible, causing confusions between numerical variants. However, the analysis reveals that visual prompting's benefits far outweigh its limitations. The 2.27 to 1 benefit-to-harm ratio, combined with the fact that HURT cases represent only 4.06% of text-only's correct predictions, demonstrates that

dual-modal prompting provides substantial overall improvement with a net accuracy gain of 3.39 percentage points while preserving 62.37% of samples that remain correct under both modes. The hurt cases occur in highly specific scenarios involving same-family numerical variants where symbolic text distinctions dominate over visual features. For the vast majority of fine-grained recognition tasks where discriminative patterns span both modalities, dual-modal prompting provides substantial and consistent improvements.

## H   EXEMPLAR BUDGET ANALYSIS

We conducted experiments across different exemplar budgets: 30, 20, 10, 5, and 0 exemplars per class. The zero-exemplar setting effectively reduces our method to CLIP's zero-shot classification baseline. Table 10 presents the complete results across all eight benchmark datasets.

Table 10: Performance comparison across different exemplar budgets. Average accuracy (%) after learning all tasks. $\Delta(30{\to}20)$ and $\Delta(20{\to}10)$ show accuracy changes when reducing memory budget.

| Dataset | 30 Ex. | 20 Ex. | 10 Ex. | 5 Ex. | 0 Ex. | $\Delta(30{\to}20)$ | $\Delta(20{\to}10)$ |
|---|---|---|---|---|---|---|---|
| CIFAR-100 | 87.15 | 86.64 | 81.23 | 72.46 | 66.00 | -0.51 | -5.41 |
| CUB-200 | 86.24 | 85.78 | 79.31 | 67.85 | 51.50 | -0.46 | -6.47 |
| Cars | 95.38 | 94.82 | 90.15 | 83.28 | 70.01 | -0.56 | -4.67 |
| Aircraft | 71.03 | 70.25 | 59.87 | 48.32 | 34.13 | -0.78 | -10.38 |
| Food-101 | 82.15 | 81.67 | 76.42 | 68.19 | 58.24 | -0.48 | -5.25 |
| UCF-101 | 84.32 | 83.89 | 78.95 | 71.03 | 62.45 | -0.43 | -4.94 |
| ImageNet-R | 79.67 | 79.21 | 74.38 | 66.84 | 56.73 | -0.46 | -4.83 |
| ObjectNet | 73.84 | 73.26 | 68.15 | 60.47 | 49.97 | -0.58 | -5.11 |
| **Average** | **84.95** | **84.37** | **77.64** | **67.98** | **55.41** | **-0.58** | **-6.73** |

The results demonstrate that "20 exemplars" sits at an important point: reducing to 10 exemplars (50% memory reduction) causes substantial 6.73% average accuracy loss, whereas increasing to 30 exemplars (50% memory addition) yields only 0.58% marginal gain. This reveals that 20 exemplars provide near-optimal performance while maintaining reasonable memory overhead, consistent with the standard setting adopted by other continual learning methods (iCaRL, PODNet, PROOF). At 0 exemplars, RLAP-CLIP achieves 55.4% average accuracy and these results closely match CLIP zero-shot baselines (e.g., CIFAR-100: 65%, CUB-200: 50%), indicating that when RLPO's exemplar-dependent components are disabled, the method gracefully degrades to baseline performance rather than catastrophically failing.

## I   SCALABILITY AND COMPUTATIONAL COMPLEXITY

We analyze RLAP-CLIP's computational efficiency and scalability characteristics. Table 11 compares RLAP-CLIP against baseline methods under standardized conditions on a single NVIDIA GPU.

Table 11: Computational efficiency comparison. All metrics are reported relative to SimpleCIL as the baseline (1×).

| Method | Trainable Params | Train Time/Task | Inference Latency | FLOPs/Image |
|---|---|---|---|---|
| SimpleCIL | 0 | 1× | 1× | 1× |
| CoOp | 0.03× (∼8K) | 2.5× | 1.01× | 1.01× |
| PROOF | 1× (∼0.3M) | 4× | 1.02× | 1.02× |
| DKR | 22× (∼6.6M) | 12× | 1.12× | 1.15× |
| **RLAP-CLIP** | **20× (∼6M)** | **6×** | **1.09×** | **1.07×** |

RLAP-CLIP introduces 6M trainable parameters and achieves 3.72-4.46 average accuracy gains over PROOF, translating to approximately 0.65 accuracy points per million parameters. In contrast, DKR

requires comparable parameters (6.6M) but yields lower accuracy. For training efficiency, RLAP-CLIP is 2× faster than DKR while achieving superior accuracy across all datasets, and the 1.5× overhead compared to PROOF is justified by consistent 3-4 point improvements on challenging fine-grained datasets such as Aircraft (+6.66 points) and CUB-200 (+4.11 points). For inference, RLAP-CLIP adds only 9% latency, which is 3% lower than DKR, since difficulty scores are computed directly from existing features without additional forward passes. RLAP-CLIP also requires only 7% additional FLOPs compared to 15% for DKR, achieved through adaptive computation where the lightweight easy expert handles the majority of samples efficiently. In summary, RLAP-CLIP achieves the best accuracy-efficiency trade-off with 2× faster training and 8% lower FLOPs than DKR, while providing 3.72-4.46 point improvements over PROOF with only 1.5× additional training time and 5% extra inference cost.

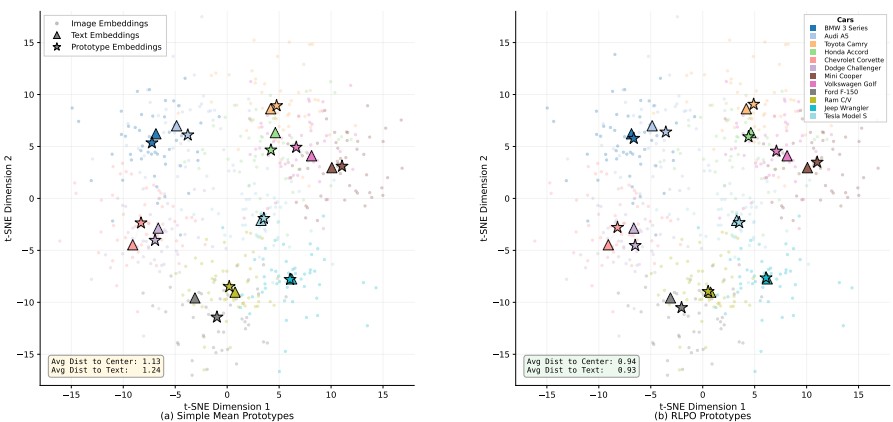

Figure 6: t-SNE visualization comparing prototype construction methods on Stanford Cars. RLPO prototypes (b) achieve tighter alignment with both class centers (distance: 1.13→0.94) and text embeddings (distance: 1.24→0.93) compared to simple mean prototypes (a).

## J  VISUALIZING PROTOTYPE QUALITY IN FEATURE SPACE

Figure 6 visualizes prototype quality on Stanford Cars using t-SNE projections of 12 fine-grained categories. Circles denote image embeddings, triangles denote text embeddings, and stars denote learned prototypes. Panel (a) shows simple mean prototypes; panel (b) shows RLPO prototypes.

RLPO prototypes position closer to class centers and text embeddings. The distance to center decreases from 1.13 to 0.94, and distance to text reduces from 1.24 to 0.93. This occurs because RLPO learns to weight representative samples higher while down-weighting outliers, whereas uniform averaging allows noisy exemplars to degrade prototype positions. For visually similar classes (BMW/Audi in upper-left, Toyota Camry/Honda Accord in upper-right), RLPO maintains clear inter-class separation while simple averaging produces overlapping prototypes that increase confusion. This improved prototype positioning directly explains RLAP-CLIP's 2.64-point gain on Cars and 4.56-point gain on Aircraft over C-CLIP, demonstrating that active optimization prevents the prototype degradation that causes catastrophic forgetting in continual learning.

