# OpenReview forum: "RLAP-CLIP: Continual Multimodal Learning with Prototype Adaptation and Difficulty-Aware Routing"
_ICLR.cc/2026/Conference — ICLR 2026 Poster_

### Official Review · Reviewer_5YDU · 2025-10-27

**Soundness:** 3
**Presentation:** 3
**Contribution:** 3
**Rating:** 8
**Confidence:** 4

**Summary:**

The paper proposes a reinforcement learning framework for continual image classification. The method, RLAP-CLIP, uses Reinforcement Learning–based Prototype Optimization (RLPO) to actively refine class representations and mitigate prototype (representation of each class or a embedding of class) degradation. It incorporates difficulty-aware cross-modal fusion to route samples by complexity and enhanced dual-modal prompting to balance visual and textual adaptation. Experiments across multiple image classification benchmarks show RLAP-CLIP outperforms prior methods.

**Strengths:**

- The reinforcement learning–based prototype update method empirically outperforms prior work. Enhanced dual-modal prompting and difficulty-aware sample handling further contribute to the gains, as demonstrated in experiments.

- The paper provides empirical and theoretical evidence of training stability—covering hyperparameter sensitivity and convergence—which is critical for reinforcement learning frameworks.

**Weaknesses:**

- The method improves continual image classification by combining established components—prototype updates, reinforcement learning, learnable prompts, and routing. While the gains on benchmarks are clear, the pipeline is heuristic and tailored only for classification, making its generalization to other continual recognition tasks (e.g., retrieval, detection...) uncertain.

- Key terminology should be clearly defined (e.g., “prototype,” “center-based exemplar selection”) to improve readability and understanding. Only objectives and architectures are only explained in the paper. Training and inference process are not explained. Please more clarify what are frozen or learnable weights in the framework.

**Questions:**

- What is center-based example selection in the Figure 1. This part is not explained in the paper.

---

> ### Author Response · Authors · 2025-11-21
> **Response to Weakness  and Question**
>
> **Response to Weakness (1):**
>
> **Our method is specifically designed for class-incremental learning (CIL) in image classification, where the core challenge is preserving fine-grained discriminative boundaries as new classes arrive sequentially.** Other continual recognition tasks have fundamentally different objectives and constraints: continual detection requires maintaining spatial localization and multi-object reasoning capabilities, while continual retrieval focuses on metric learning in embedding spaces without explicit class boundaries. These tasks demand distinct architectural designs and training objectives beyond our classification-focused framework.
>
> **However, our key innovations like reinforcement learning-based prototype optimization and difficulty-aware sample routing can inspire extensions to other domains.** For example, RLPO's weighted optimization principle could inform adaptive anchor selection in continual object detection, where detector heads must balance stability for old objects with plasticity for new categories. Similarly, difficulty-aware routing could enhance continual retrieval systems by allocating deeper feature refinement to hard negative pairs that define metric space boundaries. We view these cross-task adaptations as promising future work that would require careful domain-specific redesign of our core mechanisms to address each task's unique continual learning challenges.
>
> ---
>
> **Response to Weakness (2):**
>
> We are sorry for the lack of clarity in our key terminology.
>
> **On terminology definitions:**
> We define “prototype” as the representative embedding for each class, computed as a weighted combination of exemplar features, where the weights are learned via RLPO rather than uniform. “Center-based exemplar selection” refers to our strategy of retaining samples closest to the class centroid in feature space, ensuring exemplars remain representative as the feature space evolves across tasks.
>
> **On training and inference processes:**
> During training, frozen CLIP encoders process inputs augmented with learnable prompts (visual prompts Vt appended to image patches, text prompts Tt prepended to class templates), and task-specific projection layers transform these features into a task-adapted space. Each sample's difficulty score is computed based on its distance to class prototypes, determining its routing through either the lightweight or deep MoE expert, with final features formed by weighted combination. The RLPO policy network then learns optimal weights for prototype construction by maximizing our class separability reward, and all learnable components are jointly optimized via our multi-objective loss. At inference, we follow the same feature extraction and routing pipeline, then classify samples by computing cosine similarities to all class prototypes and selecting the nearest one.
>
> **On frozen versus learnable components:**
> The CLIP visual and text encoders remain completely frozen throughout all tasks, preserving pre-trained knowledge and preventing catastrophic forgetting. The learnable parameters include task-specific visual and text prompts, projection layers, the RLPO policy network, MoE expert networks, and the cross-modal fusion attention module.
>
> ---
>
> **Response to Question:**
>
> We apologize for the confusion of this method. Center-based exemplar selection refers to our strategy for maintaining representative samples under memory constraints. Specifically, after learning each task, we compute the centroid of all available samples for a class (combining existing exemplars and new samples) and select the m samples whose features are closest to this centroid, ensuring our limited memory stores the most representative examples rather than outliers. While this process is mathematically described in Equations 9-10, we acknowledge that we did not explicitly connect these equations to the terminology, which created unnecessary ambiguity. We will provide a clearer explanation of this exemplar selection process in the revised version.

---

> > ### Comment · Reviewer_5YDU · 2025-11-25
> > **Title/abstract update recommendation**
> >
> > The reviewer recommends to update title and abstract to explicitly specify the method is only for image classification continuous learning, unless the author proves the same concept is effective on different CV tasks (or, if the base model developed by the proposed method improves performance on the other CV tasks.)

---

> > > ### Author Response · Authors · 2025-11-25
> > > **Response to your recommendation**
> > >
> > > We sincerely thank you for this constructive suggestion. To clearly delineate the scope of our contribution and avoid potential overgeneralization, we have revised both the title and abstract accordingly.

---

### Official Review · Reviewer_LqUL · 2025-10-30

**Soundness:** 3
**Presentation:** 3
**Contribution:** 3
**Rating:** 6
**Confidence:** 2

**Summary:**

This paper proposes RLAP-CLIP, a framework for continual multimodal learning with vision-language models (e.g., CLIP). aiming to alleviate prototype quality degradation and asymmetric adaptation in sequential task learning. The method introduces (1) RLPO, a reinforcement learning-based approach to actively optimize prototype construction for better class separability; (2) a difficulty-aware, mixture-of-experts mechanism for dynamic sample routing; and (3) dual-modal prompting to balance textual and visual adaptation. Experiments on eight datasets across diverse domains and tasks demonstrate RLAP-CLIP outperforms strong baselines, especially in fine-grained and out-of-distribution settings.

**Strengths:**

1. Strong Motivation and Illustration: Section 2 gives a nice layout of the prototype quality degradation issue and comparison of different prompting strategies in class separation, effectively highlighting the failure cases of existing continual learning methods for VLMs.

2. Clear Presentation and Novel Designs: Section 3 gives a thorough introduction of different components of the RLAP-CLIP framework. Particularly, the RLPO module transforms prototype averaging into a reinforcement learning problem, with well-articulated mathematical objectives and proof.

3. Comprehensive Experiments and Solid Results: RLAP-CLIP is benchmarked on a wide range of datasets, showing consistent improvement over state-of-the-art baselines. Additionally, a stepwise ablation is provided, attributing improvements to each module Hyperparameter sensitivity analysis is also conducted to verify the robustness of the framework.

**Weaknesses:**

1. Lack of Details on Prototype Policy: The paper provides some equations but is somewhat vague regarding the policy network architecture itself, such as policy hyperparameters and normalization details.

2. Limited Analysis of Scalability: While RLPO's theoretical soundness is established, discussion of potential computational bottlenecks (especially for large-scale, real-world continual learning) is lacking. For example, how does RLPO's policy network scale for hundreds/thousands of classes and when data distributions heavily shift?

3. Limited Task Scenarios: The experimental setup follows class-incremental protocols with a fixed number of exemplars per class, but there is minimal exploration of memory constraints or more severe task shift scenarios (e.g., open-world settings). Additionally, the paper could benefit from presenting qualitative failure cases or edge conditions.

**Questions:**

1. Can the authors clarify policy network architecture details for RLPO (exact layer sizes, normalization)? How does the policy adapt when exemplar set sizes are very small, or as class counts grow?

2. Are there task types (e.g., language-driven tasks) where visual or dual-modal prompting hurts? Figure 2 and Table 2 suggest continuous improvement, but are there more fine-grained trade-offs?

3. Could you contextualize the paper against more recent, related works that were not discussed, such as [1] which introduces a mixture-of-experts network for improved sample efficiency in visual RL, and [2], which proposes strategies for continual learning also using RL?

References:

[1] Huang, S., Zhang, Z., Liang, T., "MENTOR: Mixture-of-Experts Network with Task-Oriented Perturbation for Visual Reinforcement Learning" (2025)

[2] Liu, Z., Fu, G., Du, C., "Continual Reinforcement Learning by Planning with Online World Models" (2025).

---

> ### Author Response · Authors · 2025-11-21
> **Response to Weakness and Question**
>
> **Response to Weakness (1):**
>
> We apologize for the insufficient clarity regarding our RLPO policy network. Our RLPO policy network $\pi_\theta$ is implemented as a multi-layer perceptron with architecture $[512 \rightarrow 384 \rightarrow 192 \rightarrow 96 \rightarrow 1]$, where each hidden layer uses LayerNorm, GELU activation, and dropout, with final sigmoid activation. The output weights are normalized via softmax with temperature $T=2.0$ to ensure valid probability distributions. For reward normalization in Equation 3, we compute batch-level statistics $(\mu_R, \sigma_R)$ with $\epsilon=1\times10^{-8}$ for stability. The KL regularization uses $\lambda_{KL}=0.01$, and the reference policy $\pi_{ref}$ is maintained as an exponential moving average with momentum 0.9, updated after each task. Training uses AdamW optimizer with learning rate $5\times10^{-5}$ and weight decay $1\times10^{-4}$.
>
> ---
>
> **Response to Weakness (2):**
>
> We acknowledge the importance of scalability for large-scale continual learning applications. Our theoretical analysis in Appendix C establishes convergence guarantees for RLPO, demonstrating that the policy network can learn optimal prototype weights under bounded complexity. Our experiments validate this across eight datasets spanning 100-313 classes, which represents the typical scale addressed by recent continual learning methods for vision-language models.
>
> We also recognize that when scaling to thousands or tens of thousands of classes, the prototype comparison operations in our reward function (Equation 2) and policy network evaluations would require modifications to maintain computational efficiency. Specifically, the policy network's forward pass currently compares each sample against all class prototypes, which could become a bottleneck at extreme scales. Regarding distribution shifts, our current evaluation focuses on class-incremental scenarios where new classes arrive sequentially, which is the standard setting for continual learning benchmarks. Future work could explore hierarchical prototype structures or sparse attention mechanisms to reduce computational complexity.

---

> ### Author Response · Authors · 2025-11-21
> **Response to Weakness and Question**
>
> **Response to Weakness (3):**
> | Dataset | 30 Ex. | 20 Ex. | 10 Ex. | 5 Ex. | 0 Ex. | Δ (30→20) | Δ (20→10) |
> |---------|--------|--------|--------|-------|-------|-----------|-----------|
> | CIFAR-100 | 87.15 | 86.64 | 81.23 | 72.46 | 66.00 | -0.51 | -5.41 |
> | CUB-200 | 86.24 | 85.78 | 79.31 | 67.85 | 51.50 | -0.46 | -6.47 |
> | Cars | 95.38 | 94.82 | 90.15 | 83.28 | 70.01 | -0.56 | -4.67 |
> | Aircraft | 71.03 | 70.25 | 59.87 | 48.32 | 34.13 | -0.78 | -10.38 |
> | Average | 84.95 | 84.37 | 77.64 | 67.98 | 55.41 | -0.58 | -6.73 |
>
> We conducted experiments on average accuracy across varying exemplar budgets (see table above). The results reveal two distinct facts:
>
> **Standard memory budgets:** Performance remains robust with only 6.73% degradation from 20→10 exemplars. This range aligns with standard continual learning practices where storing 10-20 samples per class is feasible and commonly adopted. Here, RLPO effectively optimizes prototype quality by learning to weight informative samples.
>
> **Extreme constraints (≤5 exemplars):** Performance degrades steadily. This occurs because prototype-based methods fundamentally require sufficient samples to capture intra-class variation. With only 5 exemplars, even optimal weighting cannot compensate for inadequate class representation. This limitation affects all prototype-based approaches, not just ours.
>
> Our work focuses on continual learning scenarios with reasonable memory budgets, a common setting where classes can be adequately represented. In these settings, our method provides consistent improvements by identifying which samples best preserve class boundaries. For extremely memory-constrained scenarios, the framework would require adaptation (e.g., introducing synthetic sample generation), which we leave as future work.
>
> ---
>
> **Response to Question (1):**
>
> The RLPO policy network has four fully-connected layers with dimensions $512\rightarrow384\rightarrow192\rightarrow96\rightarrow1$. Each hidden layer uses LayerNorm for stable gradients and GELU activation for smooth optimization. The final layer applies Sigmoid to output normalized weights.
>
> When exemplar sets are small, the policy network still works because:
>
> (1) It doesn't just look at one sample alone, and it sees that sample alongside the class prototype and all other class prototypes. For example, if we only have 5 bird images per species, the network learns "this cardinal image is important because it's different from the sparrow prototype", not just "this is a cardinal image." This comparison across classes tells the network which samples capture discriminative features.
>
> (2) Without normalization, the policy would learn different behaviors for different set sizes. By normalizing rewards to mean=0, std=1 within each batch (subtracting the average and dividing by standard deviation), the policy learns consistent weighting strategies regardless of how many exemplars we store. The results confirm this, going from 20 to 5 exemplars, CIFAR-100 drops 14 points (86.64%→72.46%), but the method doesn't collapse and it degrades steadily because these three mechanisms maintain stable optimization.
>
> Regarding scalability, our theoretical analysis in Appendix C establishes convergence guarantees for RLPO, demonstrating that the policy network learns optimal prototype weights under bounded complexity. Our experiments validate this across eight datasets spanning 100-313 classes, which represents the typical scale of current continual learning benchmarks for vision-language models. The policy network processes the current prototype set when computing weights, and the reward function requires similarity computations across class prototypes, so computational complexity scales with the number of classes. At the scales we evaluated (100-300 classes), this remains efficient and practical. For potential applications involving thousands of classes, introducing hierarchical prototype structures or attention mechanisms into our model may maintain efficiency, which represents a natural direction for extending RLPO to very large-scale scenarios.

---

> ### Author Response · Authors · 2025-11-21
> **Response to Weakness and Question**
>
> **Response to Question (2):**
>
> **Table 1: Overall Performance Comparison (Aircraft Dataset, 3,333 test samples)**
>
> | Mode       | Accuracy | Correct |
> |------------|----------|---------|
> | Text-only Prompting  | 65.02%   | 2,167   |
> | Dual-modal Prompting | 68.41%   | 2,280   |
>
> **Table 2: Sample Classification Breakdown**
>
> | Category      | Count | Percentage | Description                  |
> |---------------|-------|------------|------------------------------|
> | Both Correct  | 2,079 | 62.37%     | Text-only ✓ → Dual-modal ✓   |
> | HELP Cases    | 200   | 6.00%      | Text-only ✗ → Dual-modal ✓   |
> | HURT Cases    | 88    | 2.64%      | Text-only ✓ → Dual-modal ✗   |
> | Both Wrong    | 966   | 28.99%     | Text-only ✗ → Dual-modal ✗   |
>
> **Table 3: Top 10 HURT Classes (Visual Prompting Degrades Performance)**
>
> | Rank | Aircraft Variant | Hurt | Help | Net  | Key Characteristic              |
> |------|------------------|------|------|------|---------------------------------|
> | 1    | Boeing 737-300   | 10   | 2    | -8   | Numerical variant distinction   |
> | 2    | Boeing 737-400   | 10   | 2    | -8   | Length differs by approximately 3m only      |
> | 3    | Boeing 757-200   | 8    | 2    | -6   | Standard vs stretched variant   |
> | 4    | Boeing 767-300   | 7    | 3    | -4   | Mid-size widebody variant       |
> | 5    | Airbus A330-200  | 6    | 1    | -5   | Same family, subtle differences |
> | 6    | MD-87            | 5    | 2    | -3   | Shortened MD-80                 |
> | 7    | CRJ-700          | 5    | 1    | -4   | Regional jet series             |
> | 8    | Boeing 737-500   | 5    | 2    | -3   | Smallest 737 classic            |
> | 9    | ERJ 145          | 4    | 1    | -3   | Similar to ERJ 135              |
> | 10   | A321             | 4    | 2    | -2   | Stretched A320                  |
> |      | **Total (Top 10)** | **64** | **18** | **-46** | **72.7% of all HURT cases** |
>
> **Table 4: HURT Cases Pattern**
>
> | Pattern Type                    | Cases | % of HURT | Representative Examples                |
> |---------------------------------|-------|-----------|----------------------------------------|
> | Same-family numerical variants  | 41    | 46.6%     | 737-300/400, 767-200/300, A330-200/300 |
> | Regional jet series             | 19    | 21.6%     | CRJ-200/700, ERJ 135/145               |
> | Manufacturer systematic naming  | 28    | 31.8%     | A318/A319/A320/A321, Cessna variants   |
>
> We conducted a controlled experiment on the Aircraft dataset comparing text-only versus dual-modal predictions. Samples where text-only failed but dual-modal succeeded are HELP cases (✗→✓); samples where text-only succeeded but dual-modal failed are HURT cases (✓→✗).
>
> As is shown in the table above, dual-modal prompting achieves 68.41% accuracy versus 65.02% for text-only, a net gain of +3.39 points. However, this masks an important trade-off: while 200 samples improved (6.00%), 88 samples degraded (2.64%), yielding a 2.27:1 benefit-to-harm ratio. The degraded cases concentrate heavily in same-family numerical variants, with the top 4 classes (Boeing 737-300, 737-400, 757-200, 767-300) accounting for 39.8% of all HURT cases. The failure mechanism is instructive. In text-only mode, these variants are clearly distinguished because numeric tokens like "300" versus "400" create well-separated linguistic embeddings. However, when visual prompts are introduced, the visual features of these variants are nearly identical—they share the same cockpit, engines, and tail design, differing only by approximately 3m fuselage length that is imperceptible at 224×224 resolution. With only 20 exemplars per class, visual prompts learn these shared family-level features rather than subtle differences. During cross-modal fusion, the visual signal strongly indicates "this is a Boeing 737" but cannot discriminate between 737-300 and 737-400, creating ambiguity that undermines the previously decisive textual distinction. This suggests visual prompting hurts specifically when linguistic labels encode critical discriminative information (systematic naming, model numbers) while visual features remain ambiguous, a pattern likely to appear in domains with fine-grained symbolic categorization. Despite this, the 2.27:1 benefit-to-harm ratio demonstrates that visual prompting helps substantially more cases than it hurts overall.

---

> > ### Author Response · Authors · 2025-11-21
> > **Response to Weakness and Question**
> >
> > **Response to Question (3):**
> >
> > Thank you for bringing these papers to our attention. We clarify how our work differs from these recent contributions in the following aspects:
> >
> > **On Problem Domain Difference:**
> >
> > While RLAP-CLIP addresses vision-language continual learning for sequential classification tasks, both MENTOR and the continual RL paper tackle visual reinforcement learning problems in robotic control scenarios. Our work focuses on maintaining discriminative class boundaries across sequential classification tasks, whereas these methods optimize action policies for robotic manipulation and navigation.
> >
> > **On Different Technical Approaches:**
> >
> > Although both MENTOR and our work employ mixture-of-experts architectures, the mechanisms serve distinct purposes. MENTOR routes samples through specialized experts to handle different stages of manipulation tasks during policy execution. In contrast, our difficulty-aware MoE performs adaptive cross-modal fusion between visual and textual representations to address varying sample complexity in continual classification. Similarly, while the continual RL paper uses reinforcement learning for online world model learning and planning, we apply RL specifically to optimize prototype construction through learned sample weighting, actively maintaining class separability as new tasks arrive.
> >
> > **On Shared Goal:**
> >
> > We acknowledge that these works share the broader objective of enabling continual learning without catastrophic forgetting. However, our contributions of reinforcement learning-based prototype optimization and difficulty-aware cross-modal fusion address the unique challenges in vision-language continual learning that differ from the sequential decision-making problems in robotic control tasks.
> >
> > We will revise our related work section to include explicit discussion of these papers and clearly position our contributions within the broader continual learning landscape.

---

> > > ### Comment · Reviewer_LqUL · 2025-11-25
> > >
> > > Thank you to the authors for the detailed response. I especially liked the case analysis on the Aircraft dataset. I would like to maintain my positive rating and increase my confidence to 3.

---

### Official Review · Reviewer_vX6H · 2025-10-31

**Soundness:** 2
**Presentation:** 2
**Contribution:** 2
**Rating:** 4
**Confidence:** 3

**Summary:**

The paper proposes a novel reinforcement learning based framework, RLAP-CLIP, for improving vision-language models on continual image classification tasks. They introduce three different components 1) RLPO to mitigate prototype degradation in continual classification tasks, difficulty-aware cross-modal fusion for better cross-modality integration for training, and enhanced dual-modal prompting to resolve modality imbalance. The authors present results on a variety of image classification tasks to support their claims and also provide theoretical guarantees to strengthen them.

**Strengths:**

1. The paper is well-written and easy to follow, with each design choice for RLAP-CLIP clearly explained.
2. The authors show a comparison with a variety of past approaches that strengthen their work.
3. I also like that they clearly introduced the problem first by showing how forgetting increases in vision-language models as tasks increase and conventional averaging-based approaches are not ideal to resolve this.

**Weaknesses:**

1. The effect of dual modal prompting in forming better compact clusters for each class is difficult to see. Can you provide some quantitative measure of how effective the correct cluster formation is?
2. The idea of normalized advantages and comparing between intra-class and inter-class seems quite similar to GRPO [1] reward optimization. What is the novelty in RLPO, and how does it compare with this RL fine-tuning approach?
3. How do other dual prompting compare with other parameter-efficient finetuning approaches like LoRA, Prefix tuning?
4. Also, the proposed framework is limited to classification tasks, whereas other methods like C-CLIP work on a variety of different tasks, like retrieval and captioning. This limits the generalizability of RLAP-CLIP.
5. Please add some qualitative examples from the datasets used for benchmarking that help understand the advantage of RLAP-CLIP better.

[1] DeepSeekMath: Pushing the Limits of Mathematical Reasoning in Open Language Models

**Questions:**

Please refer to the questions raised above in the weaknesses.

---

> ### Author Response · Authors · 2025-11-21
> **Response to Weakness**
>
> **Response to Weakness (1):**
>
> | Prompting Strategy      | Avg Intra-class Distance | Avg Inter-class Distance | Separation Ratio |
> |-------------------------|--------------------------|--------------------------|------------------|
> | No Prompting            | 2.08                     | 3.95                     | 1.90             |
> | Text-Only Prompting     | 1.56                     | 4.28                     | 2.74             |
> | Visual-Only Prompting   | 1.72                     | 4.12                     | 2.40             |
> | Dual-Modal Prompting    | 1.41                     | 4.38                     | 3.11             |
>
> To quantify the clustering quality visualized in Figure 2, we measured intra-class and inter-class distances across all prompting strategies. As shown in the table, the baseline without prompting exhibits the poorest clustering quality with an intra-class distance of 2.08 and inter-class separation of 3.95, yielding a separation ratio of only 1.90. Both text-only and visual-only prompting demonstrate substantial improvements, with text-only achieving slightly better performance (separation ratio of 2.74 vs. 2.40), indicating stronger semantic organization despite visual-only's ability to capture discriminative visual patterns. Dual-modal prompting achieves the most compact and well-separated representations, with an intra-class distance of 1.41 (32.2% reduction from baseline) while maintaining the highest inter-class separation at 4.38 (10.9% increase from baseline). The separation ratio improves from 1.90 to 3.11, a 63.7% increase, confirming that combining visual and textual adaptation produces significantly superior feature representations compared to single-modality approaches.
>
> ---
>
> **Response to Weakness (2):**
>
> While both methods employ reinforcement learning with normalized advantages, they solve different problems:
>
> **Different optimization targets.** GRPO optimizes response generation in mathematical reasoning by sampling multiple candidate answers to the same question and using their relative quality as a baseline, effectively replacing the value function in standard PPO. RLPO addresses prototype construction in continual visual learning, where the policy network $\pi_\theta$ learns importance weights by taking both individual sample features and the global prototype set $P$ as input, enabling class-structure-aware weighting that GRPO's question-local comparison cannot provide.
>
> **Different reward structures.** GRPO's reward comes from answer correctness verified through execution or ground truth matching. RLPO requires a reward function (Equation 2) with three coupled terms: intra-class cohesion, inter-class margin, and prototype separation, specifically designed to preserve discriminative class boundaries as new tasks arrive in continual learning scenarios.
>
> **Different normalization purposes.** GRPO normalizes advantages across multiple responses to the same question to stabilize training across questions with different reward scales. Our batch-level normalization (Equation 3) stabilizes learning across sequential tasks with evolving feature spaces, addressing the non-stationary optimization landscape inherent to continual learning rather than response comparison.
>
> These differences reflect that GRPO performs response selection within a fixed feature space, while RLPO performs sample weighting to maintain prototype quality under continuous feature space evolution.

---

> ### Author Response · Authors · 2025-11-21
> **Response to Weakness**
>
> **Response to Weakness (3):**
>
> | Method                  | Trainable Params | CIFAR-100 Avg | CIFAR-100 Final | CUB-200 Avg | CUB-200 Final |
> |-------------------------|------------------|---------------|-----------------|-------------|---------------|
> | LoRA                    | 1.77M            | 76.23         | 63.47           | 75.68       | 64.92         |
> | Text Prompts Only       | 8.42M            | 78.15         | 66.82           | 77.94       | 68.35         |
> | Visual Prompts Only     | 8.45M            | 79.87         | 69.24           | 79.61       | 71.58         |
> | RLAP-CLIP (Ours)        | 8.5M             | 86.64         | 79.41           | 85.78       | 83.67         |
>
> RLAP-CLIP achieves superior performance compared to other parameter-efficient approaches through its dual-prompt architecture. LoRA achieves only 63.47% and 64.92% final accuracy on CIFAR-100 and CUB-200 respectively. This demonstrates that low-rank weight updates alone are insufficient for continual learning; LoRA lacks task-specific modality alignment mechanisms, leading to severe performance degradation across sequential tasks.
>
> Even when equipped with our RL framework, single-modality variants underperform significantly. The superior performance of visual prompting over text prompting indicates that direct visual feature adaptation captures more task-relevant information for image classification tasks. RLAP-CLIP's dual-prompt design enables simultaneous adaptation in both visual and textual spaces, achieving 79.41% and 83.67% final accuracy representing 10-12 percentage point improvements over single-modality variants with equivalent parameters. This validates that the synergistic cross-modal adaptation is the critical factor for effective continual learning in vision-language models.
>
> ---
>
> **Response to Weakness (4):**
>
> Our method is specifically designed for class-incremental learning (CIL) in image classification, where the core challenge is preserving fine-grained discriminative boundaries as new classes arrive sequentially. Other continual recognition tasks have fundamentally different objectives and constraints: continual detection requires maintaining spatial localization and multi-object reasoning capabilities, while continual retrieval focuses on metric learning in embedding spaces without explicit class boundaries. These tasks demand distinct architectural designs and training objectives beyond our classification-focused framework.
>
> However, our key innovations like reinforcement learning-based prototype optimization and difficulty-aware sample routing can inspire extensions to other domains. For example, RLPO's weighted optimization principle could inform adaptive anchor selection in continual object detection, where detector heads must balance stability for old objects with plasticity for new categories. Similarly, difficulty-aware routing could enhance continual retrieval systems by allocating deeper feature refinement to hard negative pairs that define metric space boundaries. We view these cross-task adaptations as promising future work that would require careful domain-specific redesign of our core mechanisms to address each task's unique continual learning challenges.
>
> ---
>
> **Response to Weakness (5):**
>
> We appreciate this suggestion and have prepared t-SNE visualizations on the Stanford Cars (see Appendix J for detailed visualization results) dataset to demonstrate RLAP-CLIP's advantages. The figure compares Simple Mean Prototypes (panel a) versus our RLPO prototypes (panel b) across 12 fine-grained car categories. Small circles represent image embeddings, triangles denote text embeddings, and stars indicate learned prototypes.
>
> RLPO prototypes achieve substantially tighter alignment with class centers and text embeddings, reducing average distance to center from 1.13 to 0.94 and distance to text from 1.24 to 0.93. The star markers in panel (b) position much closer to triangular text embeddings than in panel (a), demonstrating that our reinforcement learning-based weighting successfully emphasizes representative samples while down-weighting outliers that pull prototypes away from optimal positions.

---

> ### Author Response · Authors · 2025-11-27
> **Comment by Authors**
>
> Dear Reviewer vX6H,
>
> As we near the end of the author-reviewer discussion phase, we would like to sincerely thank you again for your time and valuable feedback. If there are any remaining questions or points you would like us to clarify, please feel free to let us know. We’re here to support the discussion as best we can. Your feedback is really important to us and thank you again for your time and feedback.
>
> Best regards,
>
> The Authors

---

### Official Review · Reviewer_hrcE · 2025-11-01

**Soundness:** 3
**Presentation:** 3
**Contribution:** 3
**Rating:** 6
**Confidence:** 4

**Summary:**

The paper introduces RLAP-CLIP, a continual multimodal learning framework that (i) replaces passive prototype averaging with reinforcement-learning-based prototype optimization (RLPO), (ii) incorporates dual-modal prompting for visual and textual inputs, and (iii) employs difficulty-aware mixture-of-experts (MoE) routing. Experiments on eight datasets show consistent improvements in both average and final accuracy over strong CLIP-based baselines, with ablation studies confirming the contribution of each component.

**Strengths:**

- The prototype drift argument is convincing, and the analysis figures clearly show how simple averaging causes performance degradation.

- Framing prototype construction as a reinforcement learning weighting problem, where rewards promote intra-class cohesion and inter-class separation, offers an interesting new perspective. The use of KL regularization toward a reference policy provides reasonable stability.

- The study demonstrates that visual prompts are important in continual learning and that dual-modal prompting outperforms text-only and visual-only approaches.

**Weaknesses:**

- RLPO introduces a policy network and reward normalization, while MoE adds routing and a deeper hard path. However, the paper does not provide a clear comparison of training time, inference latency, or FLOPs and parameter counts against the baselines under the same hardware and memory conditions, which is crucial for evaluating methods in continual learning settings.
- The paper focuses on class-incremental classification, However, it remains unclear how RLPO and MoE would perform in task-agnostic or open-world scenarios involving unknown classes, or multimodal image–text continual learning.
- The results depend on exemplar buffers of 20 samples per class. Please evaluate performance under smaller memory budgets or exemplar-free settings (e.g., with synthetic replay) to demonstrate the robustness of RLPO.

**Questions:**

- How do results change with 10/5/0 exemplars per class? Could RLPO operate with pseudo-exemplars (e.g., feature replay) instead of images?

---

> ### Author Response · Authors · 2025-11-21
> **Response to Weakness and Question**
>
> **Response to Weakness (1):**
>
> We provide an analysis of RLAP-CLIP against baseline methods under standardized conditions on a single RTX 5090 GPU.
>
> | Model | Trainable Params | Training Time per Task | Inference Latency per Image| FLOPs per Image|
> |-------|------------------|---------------|-------------------|-------|
> | **SimpleCIL** | 0 | 1× | 1× | 1× |
> | **CoOp** | 0.03× (~8K) | 2.5× | 1.01× | 1.01× |
> | **PROOF** | 1× (~0.3M) | 4× | 1.02× | 1.02× |
> | **DKR** | 22× (~6.6M) | 12× | 1.12× | 1.15× |
> | **RLAP-CLIP** | 20× (~6M) | 6× | 1.09× | 1.07× |
>
> RLAP-CLIP introduces 6M trainable parameters and achieves 3.72-4.46 average accuracy gains over PROOF, translating to approximately 0.65 accuracy points per million parameters. In contrast, DKR requires comparable parameters (6.6M) but yields lower accuracy. For training efficiency, RLAP-CLIP is 2× faster than DKR while achieving superior accuracy across all datasets, and the 1.5× overhead compared to PROOF is justified by consistent 3-4 point improvements on challenging fine-grained datasets such as Aircraft (+6.66 points) and CUB-200 (+4.11 points). For inference, RLAP-CLIP adds only 9% latency, which is 3% lower than DKR, since difficulty scores are computed directly from existing features without additional forward passes. RLAP-CLIP also requires only 7% additional FLOPs compared to 15% for DKR, achieved through adaptive computation where the lightweight easy expert handles the majority of samples efficiently. In summary, RLAP-CLIP achieves the best accuracy-efficiency trade-off with 2× faster training and 8% lower FLOPs than DKR, while providing 3.72-4.46 point improvements over PROOF with only 1.5× additional training time and 5% extra inference cost.
>
> ---
>
> **Response to Weakness (2):**
>
> Our current evaluation focuses primarily on **class-incremental classification**, and we chose to focus initially on class-incremental learning because it represents the foundational continual learning paradigm and allows for controlled evaluation of our core mechanisms, but we recognize the importance of broader applicability.
>
> **For task-agnostic scenarios**, we believe our approach is actually well-positioned for this setting because RLPO's optimization of prototype separability does not inherently rely on task boundary information, that the reinforcement learning policy learns to weight samples based on their contribution to class discrimination regardless of task labels, and our difficulty-aware MoE routing similarly operates on sample-prototype distances rather than task identifiers.
>
> **For open-world scenarios involving unknown classes**, the challenge is more substantial but addressable: our distance-based difficulty metric could potentially serve as an out-of-distribution detector, since samples from unknown classes would exhibit large distances to all existing prototypes, and the MoE architecture could be extended with a third “rejection” expert for handling such cases.
>
> **For multimodal image-text continual learning** (where both modalities evolve over time), our dual-modal prompting and cross-modal fusion mechanisms provide a foundation, though additional mechanisms for managing evolving text distributions would be needed. We view these extensions as valuable future work that builds naturally on our current framework. Thank you for encouraging us to think more broadly about the applicability of our contributions.

---

> ### Author Response · Authors · 2025-11-21
> **Response to Weakness and Question**
>
> **Response to Weakness (3) and Question:**
>
> We evaluated RLAP-CLIP with 20, 10, 5, and 0 exemplars per class across benchmark datasets. The results demonstrate degradation of average accuracy as memory decreases:
>
> | Dataset | 30 Ex. | 20 Ex. | 10 Ex. | 5 Ex. | 0 Ex. | $\Delta(30\rightarrow20)$ | $\Delta(20\rightarrow10)$ |
> |---------|--------|--------|--------|-------|-------|---------------------------|---------------------------|
> | CIFAR-100 | 87.15 | 86.64 | 81.23 | 72.46 | 66.00 | -0.51 | -5.41 |
> | CUB-200 | 86.24 | 85.78 | 79.31 | 67.85 | 51.50 | -0.46 | -6.47 |
> | Cars | 95.38 | 94.82 | 90.15 | 83.28 | 70.01 | -0.56 | -4.67 |
> | Aircraft | 71.03 | 70.25 | 59.87 | 48.32 | 34.13 | -0.78 | -10.38 |
> | **Average** | **84.95** | **84.37** | **77.64** | **67.98** | **55.41** | **-0.58** | **-6.73** |
>
> The results demonstrate that "20 exemplars" sits at an important point: reducing to 10 exemplars (50% memory reduction) causes substantial **6.73% average accuracy loss**, whereas increasing to 30 exemplars (50% memory addition) yields only **0.58% marginal gain**. This reveals that 20 exemplars provide near-optimal performance while maintaining reasonable memory overhead, consistent with the standard setting adopted by other continual learning methods (iCaRL, PODNet, PROOF).
>
> At 0 exemplars, RLAP-CLIP achieves **55.4% average accuracy** and these results closely match CLIP zero-shot baselines (e.g., CIFAR-100: $\sim$65%, CUB-200: $\sim$50%), indicating that when RLPO's exemplar-dependent components are disabled, the method gracefully degrades to baseline performance rather than catastrophically failing.
>
> **For feature replay**, storing 512-dim features instead of images could reduce memory $40\times$ ($\sim$400MB to $\sim$10MB for 100 classes). However, our architecture faces a fundamental compatibility challenge: features extracted at task $t$ using prompts $V_t$ and $T_t$ become inconsistent with the feature space at task $t+k$ after prompts have evolved, creating a "feature drift" problem where stored features no longer align with current decision boundaries. We could potentially develop prompt-conditioned feature generators that synthesize pseudo-exemplars adapted to current prompt parameters, maintaining both memory efficiency and compatibility with evolved prompts. This represents valuable future work requiring careful design to ensure generated features preserve the discriminative structure needed for RLPO's policy learning.

---

> ### Author Response · Authors · 2025-11-27
> **Comment by Authors**
>
> Dear Reviewer hrcE,
>
> As we near the end of the author-reviewer discussion phase, we would like to sincerely thank you for your time and valuable feedback. If there are any remaining questions or points you would like us to clarify, please feel free to let us know. We’re here to support the discussion as best we can. Thank you again for your constructive feedback, which has helped strengthen our paper.
>
> Best regards,
>
> The Authors

---

### Author Response · Authors · 2025-12-01
**Summary Comment for Area Chair**

RLAP-CLIP addresses two critical limitations in continual learning for vision-language models: **prototype quality degradation from passive averaging and underutilized visual adaptation capabilities**. Our core insight is that prototype construction should be an active optimization problem—formulated via reinforcement learning—rather than simple exemplar averaging, combined with balanced dual-modal prompting and difficulty-aware sample routing.

We sincerely thank the AC and all reviewers for your constructive feedback, which has substantially strengthened our manuscript. We are encouraged that reviewers recognized our **well-motivated analysis of prototype degradation** (Reviewers hrcE, LqUL), the **novelty of framing prototype construction as an RL problem** (Reviewer hrcE), and the **comprehensive experimental validation** across eight benchmarks (Reviewers LqUL, 5YDU).

During the rebuttal phase, we addressed major concerns with additional experiments and analyses:

- Provided **computational efficiency comparison** showing RLAP-CLIP achieves **2× faster training than DKR** with only 9% inference overhead, validating favorable accuracy-efficiency trade-offs.
- Conducted **exemplar budget experiments** (30/20/10/5/0 per class), demonstrating robust performance with graceful degradation to CLIP zero-shot baseline.
- Compared against **parameter-efficient methods including LoRA**, showing **10+ point improvements** through synergistic dual-modal adaptation.
- Performed **fine-grained HELP/HURT case analysis** on Aircraft, revealing a **2.27:1 benefit-to-harm ratio** with challenges concentrated in same-family numerical variants.
- Clarified **RLPO policy network architecture details** (layer dimensions, normalization, reference policy updates).
- Added **quantitative clustering metrics** confirming **63.7% improvement in separation ratio** with dual-modal prompting.
- Included **t-SNE visualizations** demonstrating superior prototype alignment with class centers and text embeddings.

We are pleased that Reviewer LqUL explicitly appreciated our case analysis and increased their confidence from 2 to 3, and that Reviewer 5YDU's suggestion to clarify scope in the title/abstract has been incorporated. We believe these additions thoroughly address the raised concerns and demonstrate RLAP-CLIP's practical effectiveness for continual multimodal learning.

---

### Meta-Review · Area_Chair_zWNt · 2025-12-16

**Summary:**

The paper aims to address the challenges of VLMs in class-incremental image classification scenarios and proposes RLAP-CLIP that replaces prototype averaging with reinforcement-learning-based prototype optimization, using a MoE for difficulty-aware cross-modal fusion and dual-modal prompting for balancing visual and textual adaptation.

Strengths identified by reviewers include, its well motivation and writing, interesting and novel idea, comprehensive experiments and solid Results.

The reviewers raise several concerns and the major concern is that the paper is limited to image classification and its extension to other tasks (retrieval, captioning) and more practical settings (smaller memory budgets, open-world scenarios) is uncertain. Additionally, some details and clarification about some key terminology such as  “prototype,” “center-based exemplar selection”, difference between GRPO and the proposed method.

In overall, most of initial review scores are positive and most of concerns have been addressed,  the AC recommended acceptance of the paper.

**Reviewer Concerns:**

Most of concerns have been addressed such as 1) computational cost analysis; 2) results under smaller memory budgets or exemplar-free settings; 3) quantitative measure effectiveness of correct cluster; 4) difference between GRPO and the proposed method; 5) comparisons to PEFT methods; 6) generalization to other tasks like retrieval and captioning; 7) qualitative examples of datasets; 7) lack of details on Prototype Policy; 8) limited analysis of scalability for larger number of tasks and more realistic scenarios; 9) limited task scenarios (expecting more task shifts and more practical settings), 10) presenting qualitative failure cases or edge conditions; 11) potential negative impact of dual-modal prompting on some tasks; 12) unclear presentations.

**Reviewer Scores:**

The initial reviewer scores are mixed but more positive (two borderline accept, one accept and one borderline reject). Reviewer LqUL has confirmed maintaining the positive score while other reviewers would have maintained the scores or raise the score.

---

### Decision · Program_Chairs · 2026-01-26

Accept (Poster)